# Plant traits poorly predict winner and loser shrub species in a warming tundra biome

Mariana García Criado [1] ✉, Isla H. Myers-Smith [1], Anne D. Bjorkman [2,3], Signe Normand[4], Anne Blach-Overgaard[4], Haydn J. D. Thomas [1], Anu Eskelinen [5,6,7], Konsta Happonen[2], Juha M. Alatalo [8], Alba Anadon-Rosell [9,10], Isabelle Aubin [11], Mariska te Beest [12,13], Katlyn R. Betway-May [14], Daan Blok [15], Allan Buras [16], Bruno E. L. Cerabolini[17], Katherine Christie [18], J. Hans C. Cornelissen[19], Bruce C. Forbes [20], Esther R. Frei [21,22,23,24], Paul Grogan [25], Luise Hermanutz [26], Robert D. Hollister [14], James Hudson[27], Maitane Iturrate-Garcia[28], Elina Kaarlejärvi [29], Michael Kleyer[30], Laurent J. Lamarque [31], Jonas J. Lembrechts[32], Esther Lévesque [31], Miska Luoto [33], Petr Macek [34], Jeremy L. May[35,36], Janet S. Prevéy[21,37], Gabriela Schaepman-Strub [38], Serge N. Sheremetiev[39], Laura Siegwart Collier[26,40], Nadejda A. Soudzilovskaia [41], Andrew Trant [42], Susanna E. Venn [43] & Anna-Maria Virkkala [33,44]

Climate change is leading to species redistributions. In the tundra biome, shrubs are generally expanding, but not all tundra shrub species will benefit from warming. Winner and loser species, and the characteristics that may determine success or failure, have not yet been fully identified. Here, we investigate whether past abundance changes, current range sizes and projected range shifts derived from species distribution models are related to plant trait values and intraspecific trait variation. We combined 17,921 trait records with observed past and modelled future distributions from 62 tundra shrub species across three continents. We found that species with greater variation in seed mass and specific leaf area had larger projected range shifts, and projected winner species had greater seed mass values. However, trait values and variation were not consistently related to current and projected ranges, nor to past abundance change. Overall, our findings indicate that abundance change and range shifts will not lead to directional modifications in shrub trait composition, since winner and loser species share relatively similar trait spaces.

The Arctic is warming up to four times the rate of the global average[1,2], resulting in reported shifts in biodiversity. In particular, the phenomenon of 'shrubification' has been extensively described across the tundra biome[3–8], with shrub species experiencing faster growth and reproduction, increases in height[9,10] and expansion into new areas[5,7,11].

Community-level trait shifts have already been observed, with taller species spreading in a warming Arctic[12]. These processes may cause reshuffling of species compositions and functional diversity, thus affecting tundra ecosystem function through biotic interactions[13–15]. Despite shrubs' dominance increase over other functional groups,

both increasing and decreasing shrub cover have been reported[16,17], and we do not yet know whether expanding and contracting shrub species share similar traits.

Species movements towards the poles and higher elevations by tracking warming temperatures have been discussed for over two decades[18–21]. Tundra species distributions are the result of long-term glacial history and inherent Arctic geography. Palaeoecological evidence indicates shrub expansion into the Arctic during the warmer Last Interglacial and the Holocene post-glacial period[22–24], signalling that rising temperatures are likely to result in further tundra shrub expansion[25]. Current range shifts are mediated by processes induced by climate change, including permafrost thaw, earlier snow melt, extended season length, increased nutrient availability and species interactions[26]; and by the amount of potential species habitat and species' colonisation capabilities that are determined by reproduction, dispersal and establishment success. Thus, climate change could favour generalist species with greater dispersal ability, reproductive rate, and competitive ability to expand into new areas[27–29]. For instance, dwarf birch (*Betula nana*) and tall willow (*Salix* sp.) species are expanding across the tundra due to their flexible colonisation strategy featuring clonal growth, and high seed dispersal capacity and rapid growth, respectively[30–33]. Thus, certain traits could most likely influence whether tundra species will expand or contract under climate change.

Plant traits have been widely used to assess species relationships with their environment[34]. As traits vary across environmental gradients, they can be indicators of plant responses to climatic conditions[35–37] and represent relevant dimensions of functional and strategic variation between plant species[38], at both species and community levels[39]. Typically, trait-based analyses use a single mean trait value per species at the global level[34], disregarding individual variability information[40–45]. Trait variation between and within populations can be markedly different[46], however, and is ultimately driven by differences among individuals, rather than between species[45]. Thus, intraspecific trait variation (ITV) might have a strong influence on ecological dynamics[41,47]. ITV accounts for 25% of total trait variation within communities, 32% among communities[45], and 23% of trait variation in tundra biome-wide data[42]. Indeed, ITV is an important component of environmental matching, and greater ITV via genetic or phenotypic variation could provide more opportunities for natural selection and adaptation[41], increasing species' chances of adapting to fluctuating environmental conditions[48,49]. Trait plasticity influences on trait community values has been assessed in site-level experiments[50], but has not considered population dynamics at the pan-Arctic scale.

Traits that are related to dispersal, colonisation and growth can provide insights into species range dynamics. Westoby (1998) defined the leaf-height-seed strategy scheme, which represents major axes of plant life history variation[51]. Plant height relates to competitive ability, with tall plants shading out shorter competitors. Specific Leaf Area (SLA) is linked to carbon investment per area of light capture, and plants with greater SLA obtain nutrients more easily. Seed mass is related to dispersal and colonisation, since lighter seeds generally travel further, though larger seeds tend to have higher germination success and seedling survival[52]. In the tundra, resource economics traits occupy much of the global trait space, while structural traits such as plant height are relatively more restricted[42,53]. Additional categorical traits with potential to influence species dynamics are dispersal mode, deciduousness, functional group and taxonomy. Since traits can explain species' responses to biotic and abiotic factors and influence their competitive ability[54,55], we would also expect traits to influence how species' distributions change in a warming climate.

Species Distribution Models (SDMs) have arisen as a flexible tool to quantify current species ranges and project their potential range shifts by combining species occurrences with geospatial information about climate variation[56,57]. However, SDMs have been criticised for their failure to incorporate evolutionary history, biotic interactions, or realistic dispersal, and assume that species are currently in equilibrium. Thus, range projections cannot fully mirror future species distributions in the same way as long-term observations could, which reflect not only changes in the environment, but also the effect of biological processes and transient species responses[58,59]. Nonetheless, SDMs still provide useful estimates of potential suitable habitat in the absence of observational data[60], and some SDMs now incorporate dispersal ability and additional parameters such as morpho-physiological traits to improve projections, thus making more realistic future predictions[38,61–65]. Since the processes of survival, reproduction, dispersal and colonisation determine a plant's range, range shifts should be associated with species' traits related to these processes. In the warming tundra biome, community composition[5,66,67] and certain size-related and resource economics traits are changing across time and space[12,42]. However, the explanatory power of plant traits on species' past cover change, their current range size and their potential for future range shifts across the Arctic remains unknown.

Biome-scale relationships between species trait values and intraspecific variation have yet to be quantified for tundra shrubs. These relationships could dictate why some shrub species are expanding/increasing (winners) while others are contracting/decreasing (losers), or showing no change. To overcome this knowledge gap, we combine species trait, range and abundance data to understand whether median trait values (MTV) and intraspecific trait variation (ITV) are associated with current range sizes in tundra shrubs. We compare two different scales of species monitoring: past changes in cover over time in monitoring plots, and biome-scale projections of species ranges using SDMs; two metrics that are generally positively related as per the abundance-range size theory[68]. We determine which categorical and continuous traits are associated with species projected from SDMs featuring dispersal rates to expand or decrease their ranges, and with species that have increased or decreased in abundance over time (Fig. 1). Considering the magnitude of observed vegetation changes in tundra ecosystems, plant traits could be a particularly relevant tool to understand range dynamics across a warming Arctic. Here, we address the following questions:

1. Can traits explain current shrub species range sizes?
   Greater height and SLA are linked to competitive ability and resource acquisition[35], and small-seeded species are associated with longer dispersal and greater seed production[52]. Thus, we expect taller shrubs with greater SLA values and lower seed mass to have the largest current range sizes. We hypothesise that greater ITV in all three traits would be positively related to species' range sizes, reflecting greater adaptations to environmental variability[41,48].

2. Do traits correspond with projected shrub range shifts and past cover change?
   Tundra plants occurring in warmer climates tend to have greater height and SLA[12,40,42]. With projected warming[1], we expect that species occupying warmer climatic niches and having more competitive strategies (greater height and SLA values) and increased dispersal capacity (small seeds) will occupy larger projected ranges and will have undergone cover increases under a warming climate, despite past abundance changes reflecting species responses in a way that projections cannot. We also hypothesise that species with greater ITV in all three traits will have greater projected ranges as they are likely to be adapted to a wider climatic niche in their current range, and thus undergo future range expansion with warming. We expect deciduous and wind-dispersed species from *Salicaceae*

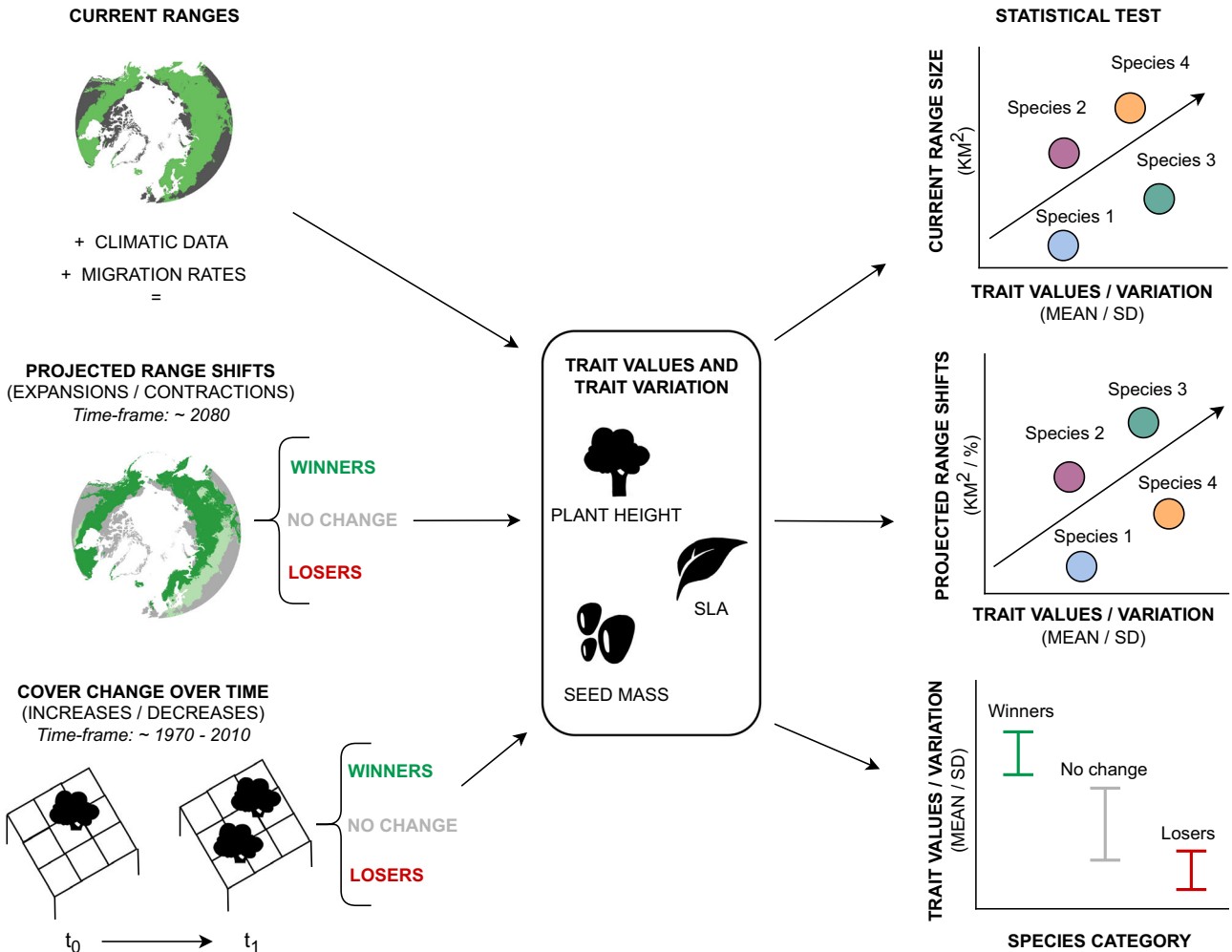

**Fig. 1 | Conceptual diagram of the different types of data used in this study and their relationships.** In the current range map, green represents the current distribution of a species. In the projected range shifts map, different green shades in the map represent the difference between current and projected ranges. In the cover change over time drawing, the point-framing grid represents cover change over time. Categories of winner, no change or loser species were identified following two different methods: based on future projections of range shifts (future winner/losers), and based on past cover change over time (past winner/losers).

Current range sizes were modelled with trait values and variation, and projected range shifts (which could be range expansions or contractions) were modelled as a function of trait values and variation. Cover change over time species categories were modelled with trait values and variation. SD means standard deviation. The polar basemaps were geo-referenced and digitized from AMAP (1998). The shrub, leaf and seed icons were commissioned for this article and designed by Alberto S. Ballesteros, who grants permission for their display here.

and *Betulaceae* families to have greater projected ranges due to their rapid resource acquisition, long-distance dispersal and flexible colonisation strategies.

3. Which are the winner and loser shrub species in a warming tundra and what are their trait combinations?

Tall plants with wind-dispersed seeds are usually more competitive as they have facilitated seed dispersal and shade shorter plants[51]. We expect winners to be mainly tall shrubs, given that they are the current dominant life form in warmer niches, and losers to be mostly dwarf shrubs, which tend to predominate in colder climatic niches[69]. We hypothesise that species with greater ITV in all traits will be winners, and vice versa for losers. Finally, we presume that species that have increased in cover (i.e., 'past winners') are also projected to experience range expansions with warming (i.e., projected 'future winners'), following the abundance-range size relationship theory[68].

Here, we show that the values and the variation in commonly recorded tundra shrub traits poorly predict species abundance and range dynamics, given that winner and loser species overlap in trait space.

## Results

Plant trait records were represented across three continents (17,921 records). SLA records were recorded for the most species ($n = 5909$ records, $n = 57$ species). Plant height records were numerous ($n = 11,466$ records, $n = 52$ species) and widespread geographically, while seed mass records were fewer ($n = 546$ records, $n = 28$ species without and $n = 40$ species with gap-filled data; Fig. 2a, Supplementary Data 2–4). By definition, there were differences in plant height values between functional groups, with tall shrubs having greater height values than low and dwarf shrubs (Fig. 2b). In contrast, most of the seed mass median values overlapped across functional groups, and the heaviest seeds belonged to dwarf shrubs (Fig. 2c). Most median SLA values also overlapped, with both the highest and the lowest median recorded for low shrubs (Fig. 2d).

### Current range sizes were not explained by traits

We did not find any clear relationships nor interactions between current range sizes and MTV (Fig. 3a–c), nor between current range sizes and ITV (Fig. 3d–f). There were no clear relationships between MTV or ITV and future winner, loser or no change categories. In contrast with

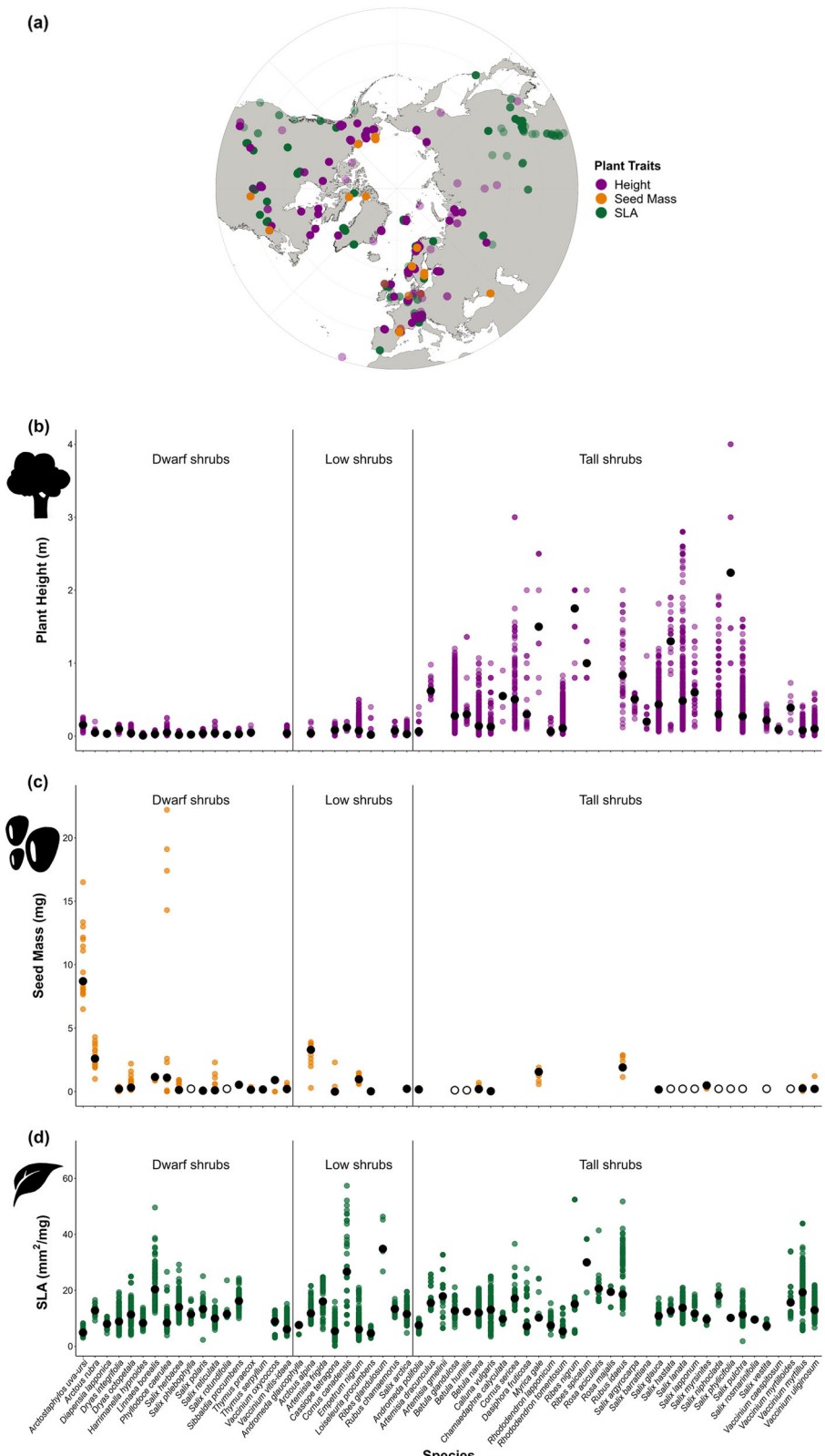

**Fig. 2 | We compiled trait data from shrubs across three continents to test whether trait values and variation were related to range size, projected range shifts and cover change.** Trait records with no coordinate information are not represented in the map. **a** Location of the geo-referenced trait records in this database, north of 30 degrees latitude. Map with polar projection created with the 'ggOceanMapsData' package, which are made with Natural Earth. **b** Plant Height values (in m) for 52 species. **c** Seed mass values (in mg) for 40 species. **d** SLA values (in mm$^2$/mg) for 57 species. Each coloured point represents an individual trait value recorded for that specific species. Coloured points are semi-transparent, with darker colour tones indicating overlaps of multiple points. Black points indicate the median value per species. Open black circles indicate the median values of seed mass for gap-filled species. Species are organised alphabetically within functional groups. The shrub, leaf and seed icons were commissioned for this article and designed by Alberto S. Ballesteros, who grants permission for their display here.

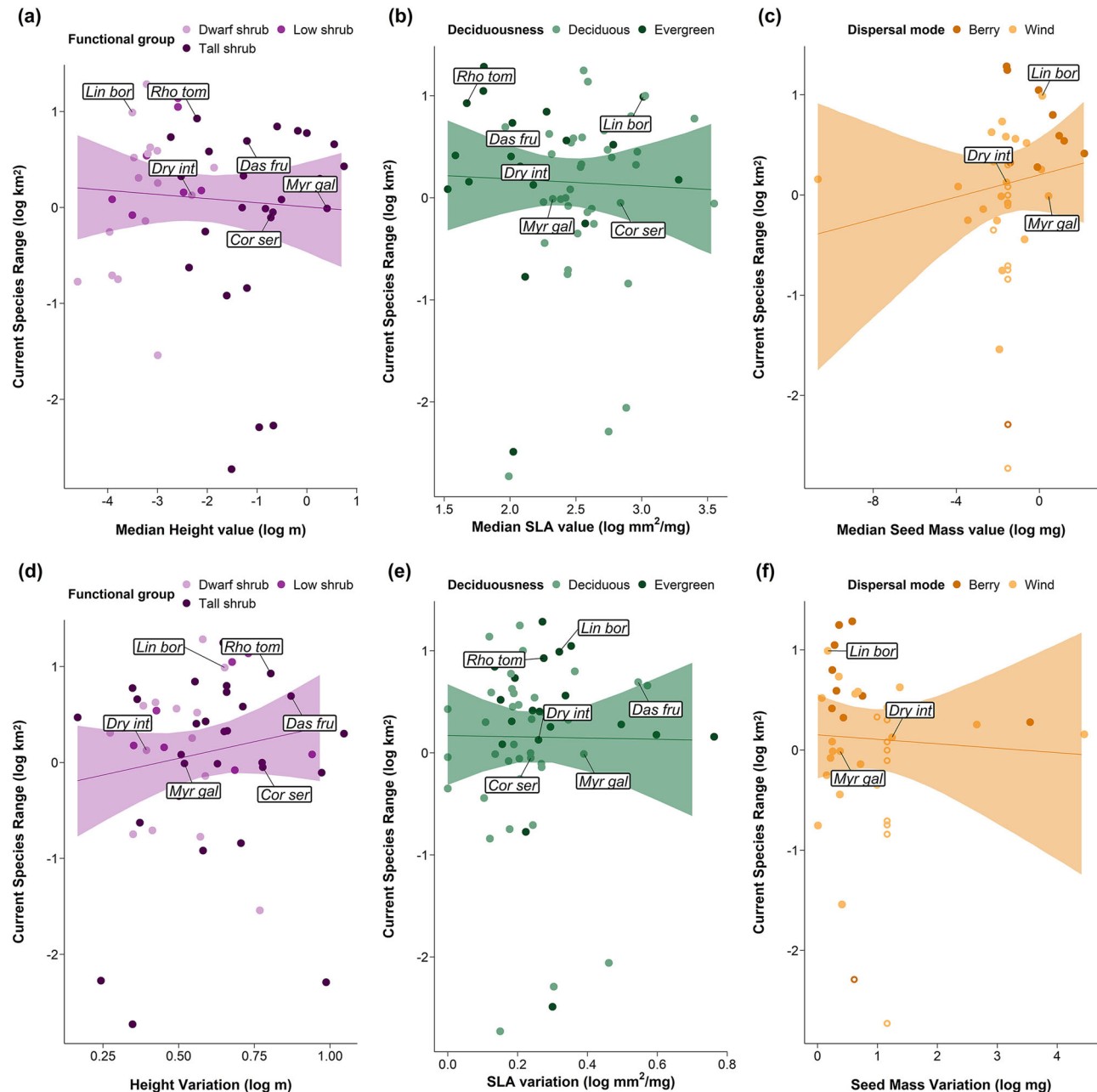

**Fig. 3 | There were no clear relationships between mean trait values (MTV) or intraspecific trait variation (ITV) and current range sizes of tundra shrubs.** Model outputs of the weighted linear regressions of current species range size as a function of **a** height values **b** SLA values, **c** seed mass values, **d** height variation, **e** SLA variation and **f** seed mass variation. MTV are the median per species and ITV is the SD of trait records. Points are coloured according to categorical traits related to each continuous trait. Lines are the predicted model slopes and the semi-transparent ribbons represent the 95% model credible intervals. Open circles in **c** and **f** represent the gap-filled seed mass points calculated from genus medians. Labels represent abbreviated species names from the top three future winners (*Rhododendron tomentosum* [previously *Ledum palustre*], *Dasiphora fruticosa* and *Myrica gale*) and the bottom three future losers (*Linnaea borealis, Cornus sericea* and *Dryas integrifolia*).

our hypotheses, there were no differences in current range sizes depending on categorical traits, including species' dispersal mode, deciduousness, functional group or taxonomic family, except for Salicaceae species having smaller ranges than species from the Rosaceae family (Supplementary Data 1.29).

**Past and future winner and loser species**
The projected range shifts method indicated similar numbers of future winner ($n = 28$, 45.2%) and loser shrub species ($n = 26$, 41.9%), and fewer no change species ($n = 8$, 12.9%) (Fig. 4). Among winner species, five were dwarf shrubs (17.9%), four were low shrubs (14.3%), and 19 were tall shrubs (67.8%). Among loser species, 11 (42.3%) were dwarf shrubs, five were low shrubs (19.2%) and 10 were tall shrubs (38.5%). For no change species, two were dwarf shrubs (25%), one was a low shrub (12.5%), and five were tall shrubs (62.5%). The winner tall shrubs were also the category-by-functional group combination with the largest number of species in this dataset. All species shared the same future winner, loser or no change category whether considering absolute (Fig. 4a) or relative range shifts (Fig. 4b).

Top winner species (of absolute range shifts) were the tall evergreen shrub *Rhododendron tomentosum,* and the tall deciduous shrubs *Dasiphora fruticosa* and *Myrica gale*. Bottom losers were the dwarf

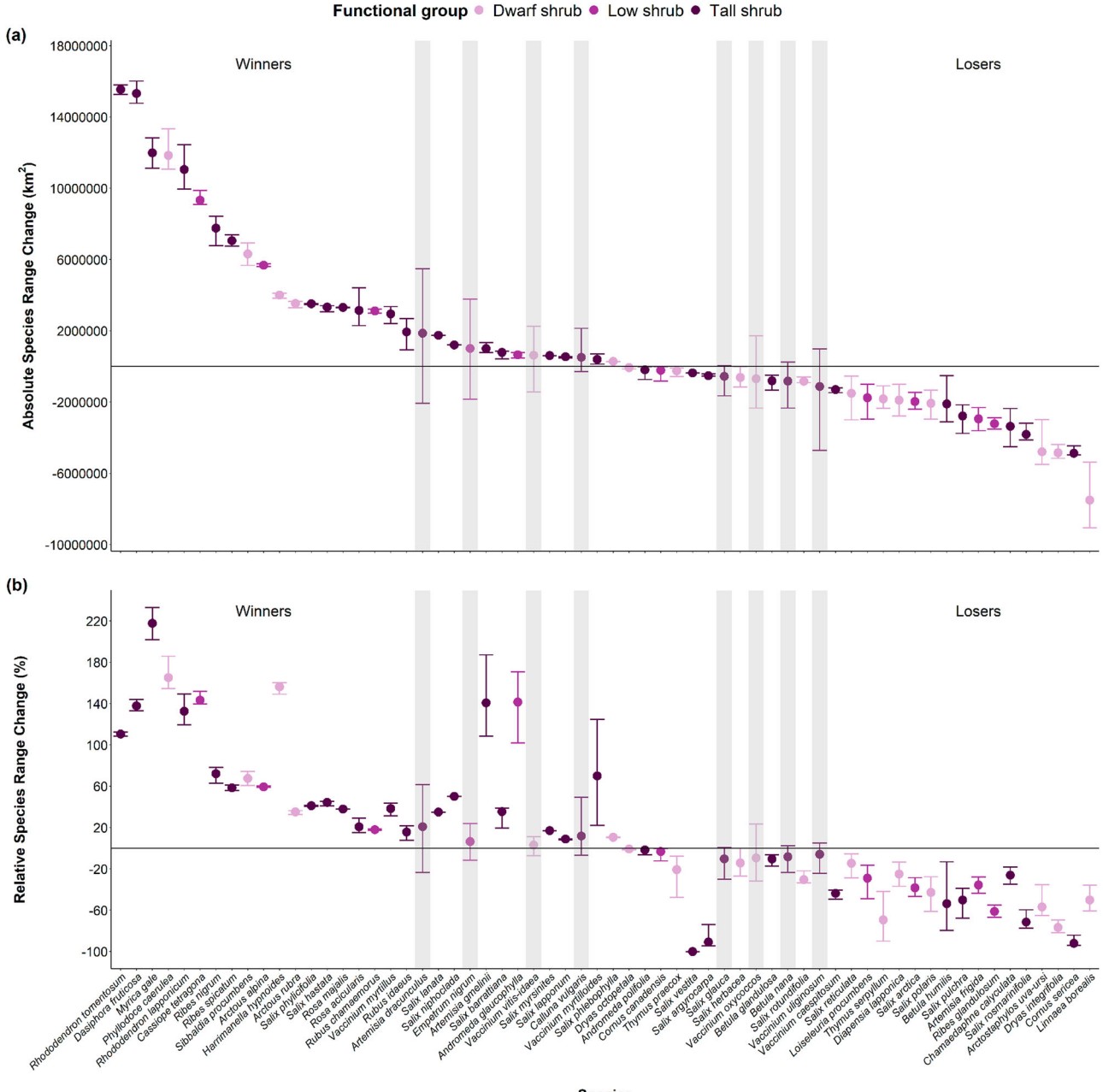

**Fig. 4 | There were similar numbers of future winner and loser species on the basis of their predicted absolute and relative species range change.** The panel shows projections of **a** absolute and **b** relative range change of tundra shrubs. Each point represents the median across the 24 predicted climatic scenarios per species, while the error bars represent the 25 and 75% quantiles of range change. Species are ordered across the horizontal axis in descending absolute change median value and coloured according to their functional group. Species whose lower quantile does not overlap zero are considered winners with expanding ranges, those whose either quantile overlaps zero are considered to experience no change (also indicated by the vertical grey polygons), and those whose upper quantile does not overlap zero are considered losers with contracting ranges. The horizontal black line represents zero range shift.

evergreen *Linnea borealis* and *Dryas integrifolia*, and the tall deciduous shrub *Cornus sericea* (Fig. 4a). Species' current range sizes and species projected range shifts were related (slope = 118.39, CI = 91.73 to 143.46), and so were median absolute range shifts and median relative range shifts (slope = 55,658.03, CI = 45,319.41 to 65,975.12). Cover change methods identified a majority of past no change species ($n$ = 19, 52.7%), nine winners (25%), and eight losers (22.2%) (Supplementary Table 2). All functional groups were represented in both past and future winner, no change and loser species. Only 10 species shared the same future and past categories (i.e., are consistently either winners, losers or no change species in both methods), with four winners, one no change and five loser species in common.

## Winners had greater variation in SLA and seed mass

Greater seed mass values were associated with greater median absolute range losses in the multivariate model (slope = −0.1, CI = −0.2 to −0.01). There was a positive interaction between height and SLA for relative median range contractions (slope = 0.6, CI = 0.02 to 1.17), with taller species with greater SLA having greater range contractions. Shrub species with greater SLA variation had greater absolute range shifts (75% quantile, slope = 0.68, CI = 0.1 to 1.25, Fig. S4a), greater relative range shifts (25%, median and 75% quantile, Fig. S4b, Supplementary Data 1), and greater relative range expansions (median, slope = 0.55, CI = 0.08 to 1.03, Fig. S4c). A 0.5 mm²/mg SLA variation increase was associated with 18 times greater projected absolute range

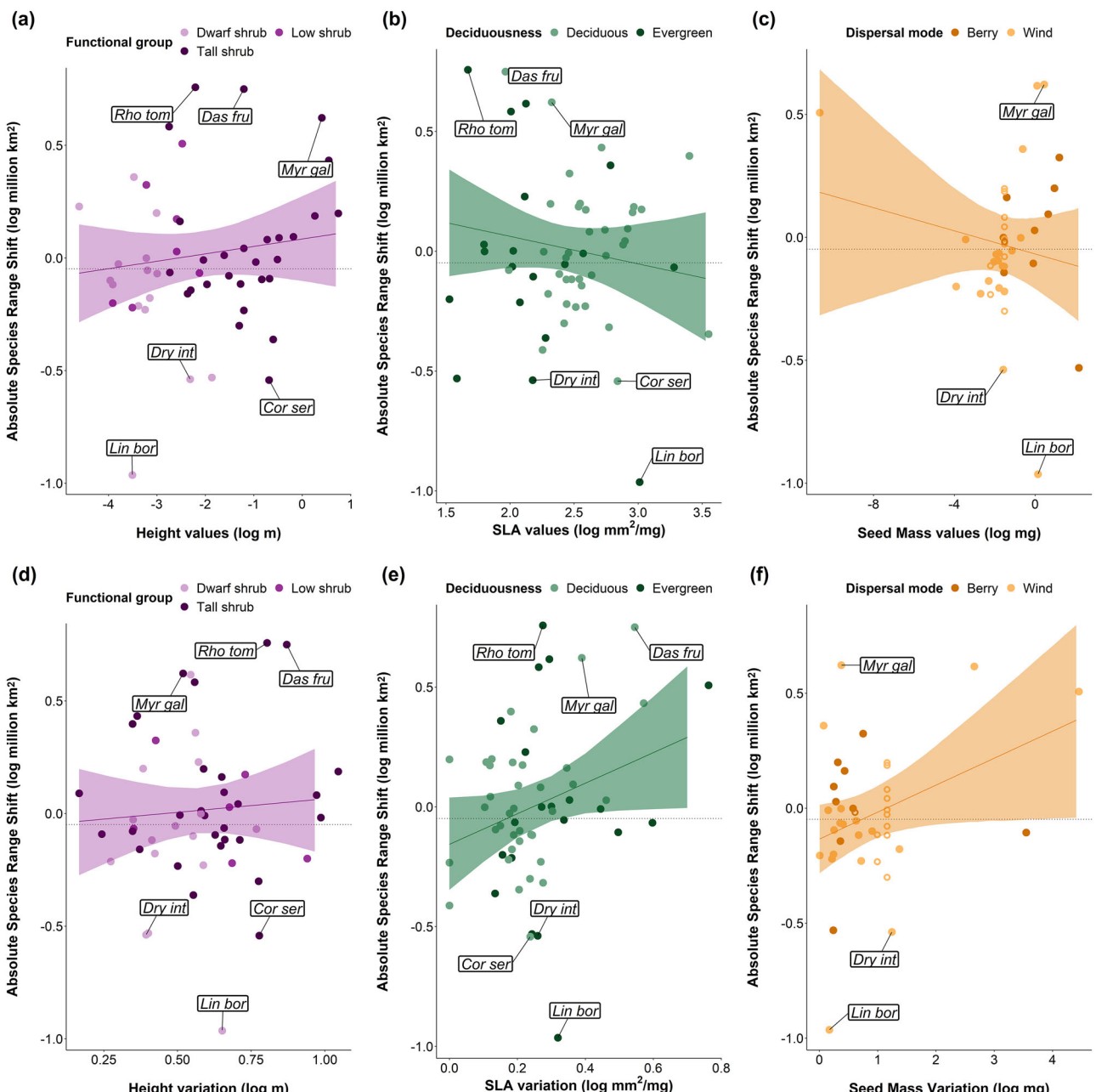

**Fig. 5 | There were no clear relationships between mean trait values (MTV) or intraspecific trait variation (ITV) and median projected range shifts of tundra shrubs, except for seed mass variation.** Model outputs of the weighted linear regressions of median absolute species range change as a function of **a** height values **b** SLA values, **c** seed mass values, **d** height variation, **e** SLA variation, and **f** seed mass variation. MTV represent the median per species and ITV is calculated as SD. Points are coloured according to categorical traits related to each continuous trait. Coloured lines are the predicted model slopes and the semi-transparent ribbons represent the 95% model credible intervals. Open circles in **c** and **f** represent the gap-filled seed mass values. Labels represent abbreviated species names as the top three future winners (*Rhododendron tomentosum, Dasiphora fruticosa,* and *Myrica gale*) and the bottom three future losers (*Linnaea borealis, Cornus sericea,* and *Dryas integrifolia*). Horizontal dotted lines indicate the zero range shift after scaling the data. Species above this line are winners and species below this line are losers.

shifts, double the relative projected range shifts and more than double the relative species expansions (Fig. S4a–c). Species with greater seed mass variation had greater absolute range shifts in univariate models (25%, median and 75% quantiles, Fig. 5f, Supplementary Data 1). This was also the case when subsetting for wind-dispersed species only (slope = 0.16, CI = 0.02 to 0.29). Note that the median absolute range shift model was only significant when including gap-filled species, but not without gap-filled species. Greater seed mass variation was related to median relative range gains (slope = 0.11, CI = 0.01 to 0.21) and absolute range gains (slope = 0.1, CI = 0.002 to 0.2). Range

expansions were ~991,273 km² larger for each mg of seed mass variation at lower values, with these relationships saturating at higher values of seed mass (Fig. S4d). We did not find any other relationships between MTV (Fig. 5a–c) or ITV (Fig. 5d, e) and median species range changes. We did not find any relationships between MTV or ITV and future winner, loser or no change category. There were no differences in projected range shifts depending on the categorical traits. However, the Caprifoliaceae family had smaller range shifts than other families, and the Myricaceae family had greater relative range shifts than Salicaceae. We did not find any clear relationships

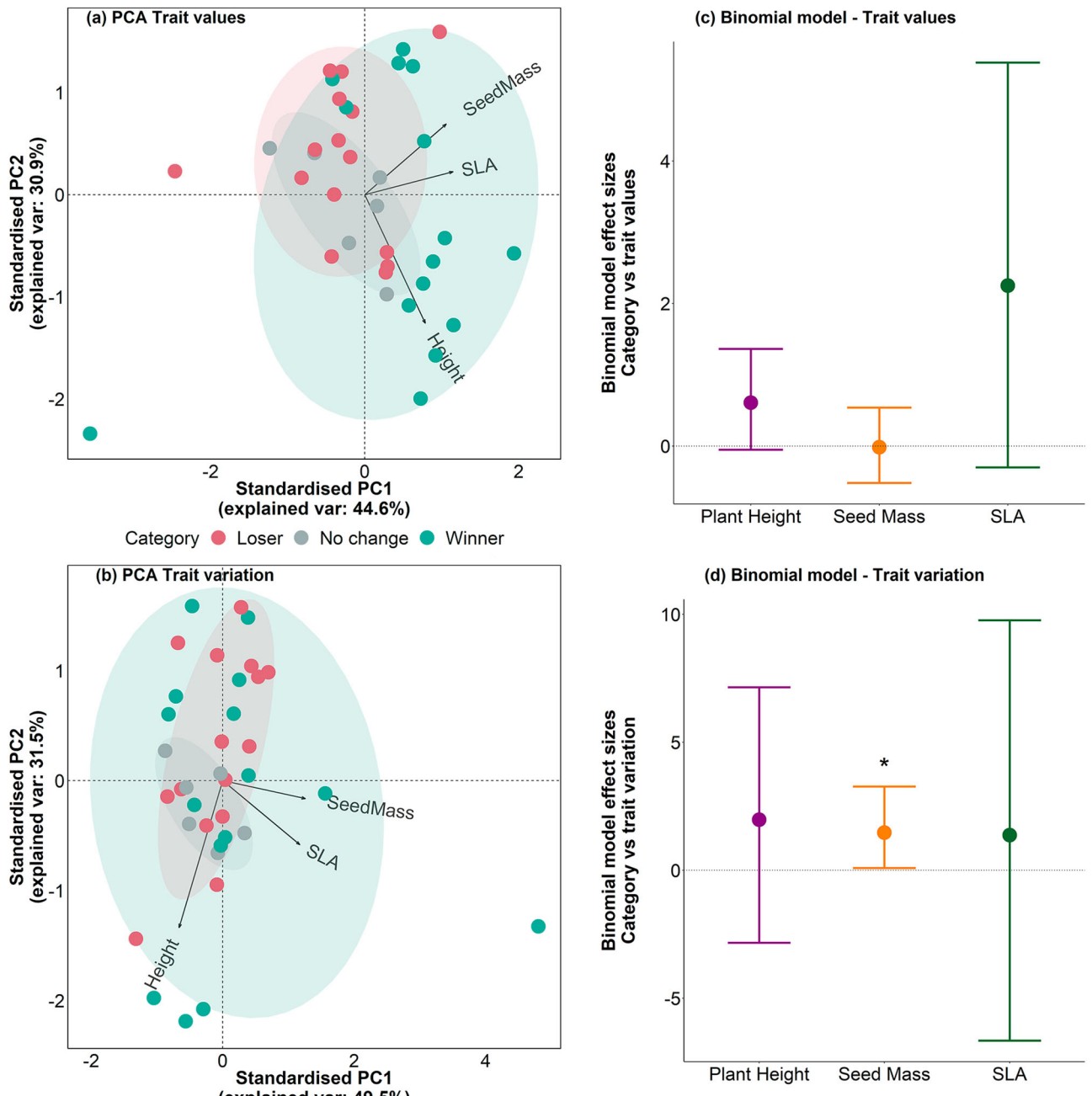

**Fig. 6 | Future winners had slightly different trait values from loser and no change species for tundra shrubs.** Principal Component Analysis for **a** mean trait values (MTV) and **b** intraspecific trait variation (ITV; $n = 36$). Ellipses and points are coloured according to species categories. Arrows indicate the direction and weighting of each trait. Ellipses indicate the 68% confidence interval of distributions per category. **c**, **d** Effect sizes of the binomial models with category (future winners versus losers and no change) as a function of **c** MTV and **d** ITV (both $n = 36$). Midpoints represent mean posterior estimates and vertical error bars represent the 95% credible intervals of the slope estimates. Asterisks indicate relationships between categories and traits that did not overlap zero (represented by the horizontal dotted line).

between the slope of average cover change over time and MTV or ITV (Supplementary Data 1.134–141).

### Winners and losers overlapped in trait space

In the MTV PCA of future categories, no change species were found across the spectrum, losers had medium to low SLA and height values, and winners had greater SLA and height values. PC1 was mainly driven by SLA and seed mass (loadings = 0.65 and 0.6, respectively), and PC2 was driven mostly by height (loading = −0.86). PC1 explained 44% and PC2 explained 30% of the dataset variation (Fig. 6a). We did not find a significant difference between groups according to the PERMANOVA

analysis ($F = 0.182$). There were no significant differences between clusters according to the pairwise comparisons for all tests and p-adjustment methods. We did not find a relationship between range categories and trait values in our binomial model, though plant height was marginally significant (Fig. 6c).

In the ITV PCA, future no change species occupied a small part of the trait space, with medium to high seed mass and SLA variation, but medium to low height variation, while losers occupied a larger part of the trait space. Future winners occupied the largest trait space for all three traits, and those species with higher variability in plant height were winners. PC1 was mainly driven by seed mass and SLA (loadings =

0.68 and 0.63, respectively), and PC2 was driven mostly by plant height (loadings = −0.91). PC1 explained 49% and PC2 explained 31% of the variation in the dataset (Fig. 6b). We did not find a significant difference among clusters in the PERMANOVA test ($F = 0.4$), but winner clusters where slightly different to no change clusters (Tukey test of multivariate dispersions, $p = 0.049$). Further, in our binomial model greater seed mass variation was more likely to correspond to winners (slope = 1.47, CI = 0.09 to 3.27; Fig. 6d). PC2 component scores had a negative relationship with relative range shifts (Supplementary Data 1.154), but we did not find any other clear relationships when modelling current range sizes, absolute range changes and relative range changes as a function of PC1 and PC2 component scores, for either MTV or ITV. We did not find differences either in winner, loser and no change categories for PC1 and PC2 scores, neither for MTV nor ITV.

Species categories based on past cover change overlapped largely in the MTV PCA, with losers having the larger trait space (Fig. S5a). PC1 was driven by SLA and seed mass (loadings = 0.7 and 0.57, respectively), while PC2 was driven mostly by plant height (loading = −0.8). PC1 explained 41% and PC2 explained 33% of the dataset variation. In the ITV PCA, clusters of past loser and winner species overlapped, though winners had greater height variation (Fig. S5b). PC1 was driven mostly by SLA and seed mass (loadings = 0.66 and 0.65, respectively), and PC2 by plant height (loading = −0.92). PC1 explained 55% and PC2 explained 29% of the dataset variation. In both PCAs, we did not find a significant difference among past winner, loser and no change clusters in the PERMANOVA test, in the binomial models, nor when modelling mean cover change over time as a function of PC1 and PC2 component scores.

## Discussion
Species' range and abundance shifts are forecasted with climate change. In this study, projected future winner species were more likely to have greater seed mass values, and greater variation in SLA and seed mass compared to losers, potentially conferring an advantage in a warmer future climate. However, the relationship of MTV and ITV with projected range shifts was highly dependent on the range shift quantiles considered per species (Supplementary Data 1). Contrary to our hypotheses, specific values of continuous traits (e.g., shorter stature) and groups within categorical traits characterised both winner and loser species. Additionally, species projected through SDMs to expand their ranges were not the same species that have increased in cover over time, showing a mismatch when employing different assessment methods. Species' projected range shifts may have consequences for the future trait composition of tundra communities[12], but not in predictable ways given that winners and losers share moderately similar trait spaces.

### Winners and losers in a warming Arctic
Plant height, SLA and seed mass are response traits that should influence species' ability to persist in and colonise changing habitats[70]. Contrary to our expectations, future winners tended to have heavier seeds than loser and no change species. Although plants with lighter seeds tend to disperse further via wind and produce more seeds[52,71], larger seeds are more likely to be found within berries that are dispersed by animals over longer distances[72] and are advantageous for seedling establishment due to greater storage tissue[52,73]. Under climatologically favourable conditions, tall shrubs and those with greater SLA have a competitive advantage over other species[74]. Tall plants may expand with increasing solar radiation and rainfall[38,75], but similar climatic conditions support communities with different MTV, and different climates can support communities with similar MTV[39]. Therefore, while macroclimate might link well with community trait values, individual trait values and ITV could instead be more affected

by microclimate, including topography, soil moisture and nutrients[76,77].

While taller species represent more future winners than shorter species (Fig. 6a), this climate-trait mismatch could mean that tall shrubs will not necessarily take over the landscape, as frequently reported in tundra projections. Surprisingly, only 10 of the 36 shrubs (27.7%) with data on past cover change over time shared the same winner/loser categories as the species range categories (Fig. 4, Supplementary Data 1, Fig. S3). This result does not support the generally accepted abundance-range size theory[68], but agrees with other studies[78]. A potential explanation is that the SDM-derived ranges identify potential future climatic niches constrained by boundaries set by species-specific migration rates, rather than the real-world climate responses of tundra shrubs. While dispersal and establishment processes are manifested in realised niches, and thus in projections to a certain extent, transient ecological dynamics are not captured by future projections. For instance, a species could be classified as a future winner because of an expanded climatic niche, but as a past loser because of decreased abundance, meaning that its fundamental niche does not track its potential future climatic niche. Conversely, a species may be classified as a loser because of a projected range contraction, but be able to persist in situ and adapt to changing climatic conditions, which SDM projections would not be able to capture.

The environmental factors affecting broad geographical extents likely differ from those affecting local-scale abundances[78]. Additionally, range shifts are contingent on geographical context, and species responses might differ depending on the space available for expansion (e.g., in North America versus Scandinavia). Moreover, biotic interactions (e.g., competition, herbivory) at local scales dictate the realisation of potential climatic niches[17,79]. Topography also influences plant growing conditions through numerous geological and hydrological processes and has been shown to improve SDM predictive ability[14]. This complexity highlights the challenges in estimating plant responses to warming where abundance increases may not translate directly into range expansions derived from SDM approaches.

### Plant traits were not strongly related to shrub species ranges or abundance
While traits have been extensively linked to predicting range dynamics and ecosystem function[80,81], we found that the traits used in this study were weakly related to the projected range shifts and past cover change of tundra shrubs (Fig. 5). Previous studies have yielded similar results. Habitat availability was more relevant than selected traits as range shift predictors for Swiss alpine plants[28], seed mass or plant height and area were not related in herbaceous plants in Swedish forests[82], and neither seed mass nor plant height predicted current species ranges for European plants[83]. Likewise, a global meta-analysis and a systematic review found no significant effect of traits (apart from habitat breadth and historic range limit) on range shifts[71,84]. Moreover, there is some evidence of poor trait predictive ability of long-term ecosystem properties, plant-environment relationships, and vital rates[85,86]. Altogether, these results indicate that contrary to our hypotheses and previous studies[87,88], these three key plant traits are not consistently associated with projected climate-induced range shifts.

Range shifts and cover change were also not defined by categorical traits such as taxonomy, dispersal mode, deciduousness or functional group (Supplementary Data 1). We expected the *Salicaceae* or *Betulaceae* families to be the greatest winners given their reported increases across tundra ecosystems[12,32,66], but our family sample size was potentially too small to detect a taxonomic signal. Although wind-dispersed (anemochorous) seeds generally have greater migration rates than animal-dispersed (zoochorous) seeds[89], we did not find anemochorous species to have larger current or projected ranges than

zoochorous species. Thus, both wind and animal dispersion might facilitate long-distance dispersal, or other factors like vegetative propagation or seed viability might be more relevant in explaining dispersal[54]. Our analysis also showed no deciduousness-related differences. Tundra deciduous shrub species are expanding with warming likely because of more efficient resource acquisition from rapid leaf turnover[66,90], but evergreen shrubs have also been responsive to warming[91–93]. All tundra plant functional groups are expected to be represented in a warming tundra[94] with large overlaps in trait values and variation between groups[40,43]. We found an indication that species projected to expand the most had greater SLA and seed mass variation, suggesting that winner species could be more plastic or adaptable[41]. However, these results were far from consistent and support the general finding that tundra species will have highly individualistic and heterogeneous responses to climate change[43,95–97].

## Moving beyond functional traits

Our initial hypothesis, based on previous literature, of tundra shrubs showing consistent trait responses to climate change turned out to be too simplistic. This weak relationship between shrub species' traits and ranges could be explained by a number of factors. First, the species' projected range shifts might be related to dispersal and colonisation processes not captured by the selected traits. Therefore, a different suite of morpho-physiological traits underpinning climatic preferences might have more explanatory power, such as leaf, stem and root density, C, N, and P contents, and cold hardening[35,98,99]. Similarly, different traits might be more related to local abundance than to projected range shifts. Second, SDM range projections may only quantify part of the full species climatic niche due to the limitations in predictor data (e.g., uncertainties in climate predictions, lack of microclimatic data) and to potential bias in the input occurrence data caused by sampling bias and long-term dispersal limitations[100]. Third, SDM projections were constrained by species-specific migration rates to avoid overestimating range shifts, but uncertainties remain regarding the influence of biotic interactions on future range shifts[59,101–103]. However, in the absence of long-term monitoring studies of traits and range shifts over time, SDM-derived projections are the best spatial data currently available to test these questions.

The filtering role of demographic processes such as survival, fecundity, germination and establishment might affect range shifts more than traits per se[104,105]. Demographic processes might be more relevant than dispersal in the tundra given the substantial role of microclimate in defining species reproduction, but they are much harder to measure than traits[106]. Although long-distance colonisation is common in the Arctic, multiple successful recruitment events are needed for a species to expand into a new area[27]. Establishment might limit distributions more than dispersal, with establishment being in turn determined by the number of viable seeds and the environment[107]. Both environmental conditions and biotic interactions such as herbivory and both intra- and inter-specific competition can heavily affect demography[101,108]. Further research is needed to understand if demographic rates could prove to be more powerful predictors of climate change-induced range shifts than dispersal traits[86,104].

We worked under the assumption that MTV and ITV will remain constant over time, but there is an indication that plant height, leaf area and seed mass will change with climate change[12,109]. With repeated tundra trait data being rarely collected over time[12], we included species records outside of the tundra to account for trait plasticity and the likelihood of tundra trait values shifting in the future[42]. Arctic geographical coverage in TRY/TTT is also incomplete (Fig. 2a) potentially leading to an under-representation of rare species. Further trait data collection across the tundra biome over time would enable the replication of these analyses based on a larger number of morpho-physiological traits and species.

With range change data over time not yet available, SDM projections remain currently the only way to estimate range dynamics. Projections provide a proxy for potential range shifts, and the relationships we found partly reflect the assumptions made when calculating SDMs. These SDMs did not consider other environmental variables beyond temperature and precipitation, and we found strong differences in projected range shifts between the 24 different climatic scenarios. Once range change data over time becomes available across tundra regions (e.g., through the GLORIA and MIREN networks[110,111]), the relationship between observed range shifts and traits could be further explored, and SDMs can be validated against on-the-ground observations. Additionally, analysing different metrics of trait variation renders an interesting research avenue but requires larger sample networks to ensure robustness of results, and thus should be considered in future studies.

Earth system models (ESMs) assume high uncertainty[112] and usually simplify diverse plant communities using functional types parametrised with summary trait values[113,114]. While acknowledging that moving beyond broad functional types will increase model complexity[113], we advocate for ESMs to incorporate trait variability and demographic processes. Progress is already underway through the definition of Arctic-specific functional groups and the inclusion of certain traits on Earth Land Models, improving overall projections[115]. In order to more accurately project tundra vegetation shifts, incorporating the real-world complexity inherent in the diverse tundra shrub responses to a warming climate remains crucial.

Our findings indicate that no specific combination of trait values or variation is associated with winner or loser tundra shrub species under climate change. Contrary to our expectations, particular trait values or greater trait variation do not necessarily indicate increased range or abundance shifts, although there was a broadly positive signal of greater seed mass values with projected range shifts, and greater SLA and seed mass variation with projected range shifts. Overall, we observed similar values of height, SLA and seed mass for both range expanding and contracting tundra shrub species. Thus, projected range shifts will not lead to directional shifts in shrub trait composition or variation, as both winner and loser species share relatively similar trait spaces. Additionally, winner and loser species differ when comparing past cover change over time with future projected range shifts. Future research could investigate the explanatory power of other morpho-physiological traits and address how demographic processes might mediate tundra shrub range shifts. Our results demonstrate that tundra shrubs can be equally resilient or vulnerable even with very different combinations of trait values and variation. Identifying the future winners and losers of climate change in the tundra biome remains a complex endeavour, but these results outline that the wide variety of evolutionary strategies that tundra plants employ are not necessarily reflected in their responses to a warming climate.

## Methods

### Definitions and taxonomy

The tundra is defined as the region beyond the elevational and latitudinal treeline[116]. We consider shrubs as multi-stemmed woody plants under 5–6 m in height[117]. We followed the taxonomy outlined in The Plant List (http://www.theplantlist.org/) at the species level and standardised synonyms according to this reference. Definitions of the three traits follow Kattge et al.[118] which in turn follow Garnier et al.[119].

### Trait data

We extracted a total of 17,921 trait records from the TRY 5.0[118] and the Tundra Trait Team (TTT) databases[120] for three plant size and economics traits related to competitive ability and dispersal (plant height, SLA and seed mass) for 62 shrub species across three continents (Fig. 2, Supplementary Data 2–4). See Supplementary Information for a list of the published datasets within the downloaded TRY database.

From the total, three trait records were from the literature and 192 records were collected by the authors and unpublished thus far. We removed the observations with values greater than four standard deviations from each species mean following the protocol outlined in Bjorkman et al. (2018)[12]. Functional traits have been correlated to each other in the literature[38,73,82], but we did not find correlations between the traits in this dataset that might have influenced our statistical outcomes (Supplementary Data 1.17–19).

We retained all georeferenced records above 30 degrees north in latitude, as we were interested in trait variation per species beyond tundra biome values. Trait data from more southern latitudes could be indicative of the trait changes that tundra species could experience in a warmer future due to adaptation, phenotypic plasticity or gene flow[42,120]. We included non-georeferenced trait records from databases that we were certain contained records from high-latitude ecosystems (e.g., if an approximate location/site name was provided). We retained only records that reported single values and individual means. We kept control and ambient values only and removed all experimental treatments and herbarium specimens as we were interested in traits from unmanipulated wild specimens. For each species-by-trait combination, we only retained those with more than four records, providing a dataset with 62 species.

We calculated 'trait values' (MTV) as the median per species and 'trait variation' (ITV) as the standard deviation (SD) of all trait values per species (Fig. 1). We chose SD as a commonly used ITV metric with a more conservative data distribution than others like the coefficient of variation (COV); however both metrics were directly proportional (Supplementary Data 2–4, Fig. S6). We compared ITV values using a random sample of five records versus all available records and found very similar data distributions, thus we opted for including all available records for ITV calculation (Supplementary Data 2–4). We log-transformed the median and SD values with the natural logarithm because the differences between species are better characterised on a log-scale[12,42,51].

To explore the influence of categorical traits, we obtained data on taxonomic family, functional group, dispersal mode and deciduousness from a variety of sources including TRY and online florae combined with expert knowledge (see 'Online sources of categorical traits and maximum height' in Supplementary Information). To group species according to height, we extracted the potential maximum canopy height per species from online florae (see Supplementary Information) and assigned the species a category following the classification in Myers-Smith et al.[121]: dwarf shrubs (<20 cm), low shrubs (20–50 cm), and tall shrubs (>50 cm). Maximum canopy height is a relevant method to classify species given our interest in the height that species could achieve in warming conditions (i.e., current height of species at sites outside the Arctic), rather than its average representative height in the Arctic. We used values from online florae rather than TRY/TTT values to avoid circularity in defining functional groups. This could mean that online florae values (mostly from the Arctic) would reflect shorter values than TRY/TTT (which include records outside the Arctic).

When screening identified duplicate records per species, trait, coordinates and collector/databases, we consulted the original datasets (when available) to investigate if potential duplicates were actual values. If both values (i.e., including duplicates) appeared in the original dataset, they were considered valid records. We removed records that were clearly duplicates (n = 129), either because they were found both in TRY and TTT, or because the original database showed no duplicates. We identified two mistakes in trait units or coordinates, which we double-checked with the original data contributors and corrected accordingly.

Since we only had original seed mass data for 28 species (as opposed to 57 species for SLA and 52 species for height), we gap-filled seed mass data for an additional 12 species that had data on both height and SLA but no seed mass data. To gap-fill, we extracted data at the genus level above 60 degrees north (to ensure Arctic representative records) and for which there were records for over four individuals. We then calculated the log-transformed median value and the SD at the genus level and included these 12 values for the gap-filled species (Supplementary Table 1).

To account for confidence depending on the number of observations, we calculated an index value per species-by-trait combination. Species with over 20 observations were assigned an index value of 1 and gap-filled species or those with five observations had an index value of 0.5. For species with between 6 and 19 observations, we calculated the index following a linear regression (see below), where $N_{obs}$ is the number of observations per species-by-trait combination:

$$Index = 0.33 + \left(\frac{1}{30}\right) * N_{obs} \quad (1)$$

We used this index to down-weight species with fewer records in the weighted regressions explained below[122,123]. We also calculated a combined index per species by averaging the individual trait indices together.

### Range size data

We used projected current range sizes to represent present-day species ranges (see Supplementary Information). To characterise projected shifts in species range size (hereafter 'range shifts'), we used SDM-derived distribution data for 62 species under 24 future climatic scenarios (see 'Species distribution modelling' in Supplementary Information for details). Three scenarios were calculated: a 'no dispersal', a 'limited dispersal' and an 'unlimited dispersal' scenarios. A 'limited dispersal' accounts for species-specific future migration rates, which were calculated using species-specific dispersal capacities in a linear mixed models framework following Tamme et al.[124], and estimate how far a species can disperse using dispersal-related traits including plant height and seed size in order to quantify more ecologically relevant range shifts. A 'limited dispersal' scenario incorporates geographical constraints, while an 'unlimited dispersal' climatic scenario (without dispersal rates) means that species in one continent could spread to another, e.g., North American species would have available ranges in Europe, and vice versa. Thus, 'unlimited dispersal' scenarios do not consider geographical realities and would likely over-estimate range sizes. We compared the three dispersal scenarios and concluded that a 'limited dispersal' scenario would be the most realistic and thus chose this scenario as representative of range shifts (Fig. S1). We also determined that the potential circularity on using the 'limited dispersal' scenario does not influence the main findings of this study (see 'Use of traits in model projections' in Supplementary Information, Fig. 5, Fig. S2, Supplementary Data 1).

Projected species range shifts were computed both as relative (%) and absolute (km²), and 'range shifts' only reflect a change in the overall range size over time (not changes in the shape or location of the ranges). We refer to 'range shifts' when we include both projected increases and decreases in total range size, and to 'range expansions' and 'range contractions' when referring to projected range size increases and decreases, respectively (Fig. 1). We log-transformed (with the natural logarithm) and centred current range sizes as the values were not always normally distributed and included outliers. Since the projected range shift data included negative values, we first divided the absolute range changes by a million km² and the relative range changes by 100, in order to bring the values closer to zero. We then added a constant value (the negative minimum value plus one) so all values were positive, and afterwards log-transformed these values. Finally, we centred these values on zero before carrying out the statistical analysis in order to facilitate convergence[125,126].

## Classification of winner and loser shrub species

We classified winner and loser shrub species using (1) projected range shifts from the SDMs (into the years 2070–2099) and (2) cover change over time from the International Tundra Experiment (ITEX) dataset (between 1970 and 2010). For range shift projections, we calculated the 25%, 50%, and 75% quantiles of species' projected range shifts across the 24 climatic scenarios (both for absolute and relative range shifts) and categorised species as projected 'future winners' (if the 25% quantile was above zero), no change (if any quantile overlapped zero) or projected 'future losers' (if the 75% quantile was below zero). For cover change over time, we analysed shrub cover change over time from 105 subsites and 30 sites from the ITEX network[127]. Based on the analysis by Bjorkman et al.[12], individual species' relative cover change over time per plot were modelled as ordinal numbers using a Poisson distribution with subsite and site as random effects, aggregating after to subsite and species level. Thus, we obtained slopes of cover change per year for each species-by-site combination. We defined past winner, no change and loser categories according to whether these slopes per species across all sites were positive or negative, and whether the 95% credible intervals overlapped zero (Fig. S3, Supplementary Table 2).

## Statistical models: Current range sizes and traits

To understand whether species' ranges were associated with traits, we fitted weighted linear regressions per trait of species' current range sizes as a function of MTV, weighting each record according to the scoring index described above. We also modelled current range size as a function of the three traits' MTV together for those species which had trait data for all three traits (weighting according to the combined index), and as a function of three two-way interactions of these three traits. To evaluate whether range size was explained by categorical traits, we modelled current range size as a function of deciduousness (evergreen/deciduous), functional group (tall/low/ dwarf shrub), dispersal mode (berry/wind-dispersed) and taxonomic family. We modelled MTV as a function of species' range category (winner, no change, loser) per trait to identify differences in trait values between the different species categories. We also modelled categories as a function of all three different traits to understand whether winners differed in their trait combinations from loser and no change species (as this was indicated in the PCA analysis described below). To do this, we fitted an additive weighted binomial model with a Bernoulli distribution by assigning a value of 0 to loser and no change species, and a value of 1 to winners. We do not include here the variant of that model with an interaction since the model did not converge with that level of complexity. Finally, we fitted similar weighted regressions as described above, with the same structure but with ITV instead of MTV (Fig. 1, Supplementary Data 1).

## Statistical models: Species range shifts and traits

To understand if species' range shifts were associated with traits, we fitted weighted linear regressions of relative and absolute range change as a function of MTV per trait, each with the 25%, 50%, and 75% quantile range change (of the 24 climatic scenarios) as a response variable. We also modelled both median relative and absolute range shifts as a function of all three traits (using the combined weighting index), and as a function of their three two-way interactions. To evaluate whether range shifts were explained by categorical traits, we fitted separate models for absolute and relative range shifts as a function of deciduousness, functional group, dispersal mode and family. To understand the processes of range expansion and contraction separately, we fitted weighted regressions for species that are predicted to experience range 'gains' and 'losses' (defined as those species whose median range change was above and below zero, respectively, both for absolute and relative

changes). We modelled median range 'gains' and 'losses' as a function of trait values per individual trait, and then as a full model with all three different traits, and their three two-way interactions, for absolute and relative changes. We fitted similar weighted regressions as described above, with the same structure but with ITV instead of MTV (Fig. 1, Supplementary Data 1). Finally, to understand if traits were related to past cover change, we fitted weighted linear regressions of the slopes of cover change over time (1970–2010) as a function of MTV and ITV, and modelled the slopes of cover change over time as a function of all three traits (Supplementary Data 1). We also modelled cover categories (winners, no change or losers) as a function of the three traits' MTV and ITV in an additive weighted binomial model with a Bernoulli distribution similarly to above.

## Statistical models: Distribution models

To understand whether species' absolute and relative ranges were related, we fitted a linear model of absolute versus relative range shifts. To investigate whether species with a larger current range were projected to expand more, we modelled future ranges as a function of current range sizes. We also fitted weighted linear regressions of current range sizes as a function of category (winner, no change or loser), and with median range change (both absolute and relative) as a function of category per individual trait to understand whether species' trajectories were related to smaller or larger present and future ranges (Supplementary Data 1).

## Statistical models: Ordinations and analyses of variance

To identify differences between species groups, we performed two Principal Component Analyses (PCAs): one for MTV and another for ITV, using the 'prcomp' function in the R 'stats' package. We centred and scaled log-transformed trait values prior to computing the PCA. We used the R package 'AMR'[128] to visualize the trait space for the 36 species for which we had data available on all three traits (including gap-filled species), and plotted the first two component axes. We extracted the PCA scores per species and used them as response variables in linear models against current range sizes, absolute and relative range shifts, and cover change slopes, and we modelled individual PCA scores as a function of winner, loser or no change range category, both for MTV and ITV, and for range and cover species categories.

We performed a permutational multivariate ANOVA test (PERMANOVA) to determine if the different groups (winners, no change or losers) differed statistically in trait space, both for MTV and ITV. We used the 'adonis' function in the R package 'vegan'[129] and specified Euclidian distance with 999 permutations. We also calculated average distance to centroids per group with the 'betadisper' function in 'vegan', and performed an ANOVA test to confirm homogeneity of dispersion among the groups ($p > 0.05$). When the 'adonis' analysis yielded a significant difference between categories ($p < 0.05$), we performed pairwise comparisons between them for 999 permutations and fitted the tests of Pillai, Wilks, Hotelling-Lawley, Roy and Spherical, and specified different methods for $p$-value adjustment, including Holm and Bonferroni, and with no $p$-value adjustment. All tests yielded similar significance results. We followed the same methods outlined above for the range and the cover change categories.

## Software and model specifications

We used the software and programming language R version 3.6.2[130] for all analyses. We fitted all Bayesian models using the 'brms' package[123] and ran them for as many iterations as necessary to achieve convergence, which we assessed through examination of the $R_{hat}$ term and trace plots. We considered that there was a clear relationship between variables when the 95% credible intervals of the estimates did not overlap with zero.

**Reporting summary**

Further information on research design is available in the Nature Portfolio Reporting Summary linked to this article.

## Data availability

Trait data are available at https://www.try-db.org/TryWeb/Home.php (TRY) and https://tundratraitteam.github.io/ (TTT). Cover change over time data will be published at https://github.com/annebj/ITEX30_VegComp. A previous version of this dataset can be accessed at http://polardata.ca/, CCIN Reference Number 10786. Species range data are available as a summarised dataset, together with the rest of input data necessary to reproduce figures and analyses, at https://zenodo.org/record/7974602. DOI for this dataset is https://doi.org/10.5281/zenodo.7974602.

## Code availability

The R code to generate the figures and analyses of this manuscript is accessible at https://zenodo.org/record/7974602.

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

## Acknowledgements

We thank Alberto S. Ballesteros for fixing picture issues and designing the shrub, leaf, and seed icons. We thank all tundra data collectors and supporting organisations, including members of the International Tundra Experiment Network (ITEX) for their efforts in data collection and for making their data accessible. We are grateful to all trait data collectors who made their data available through the TRY and TTT databases. We thank local and Indigenous peoples for the opportunity to work with data collected on their lands. M.G.C. was supported by the Principal's Career Development PhD Scholarship from The University of Edinburgh, the Elizabeth Sinclair Irvine Bequest and Centenary Agroforestry 89 Fund, and the BritishSpanish Society Award. I.H.M.-S. was supported by the NERC Shrub Tundra grant (NE/M016323/1). A.D.B. was supported by the Knut and Alice Wallenberg Foundation (Wallenberg Academy Fellowship 2019). A.B.-O. was supported by VILLUM FONDEN's Young Investigator Programme (VKR023456 to S.N.), and The Danish Council for Independent Research: Natural Sciences (DFF 4181-00565 to S.N.). B.C.F. was supported by Academy of Finland decision no. 256991, European Commission Research and Innovation Action no. 869471, and JPI-Climate no. 291581. G.S.-S. was supported by the University Research Priority Program on Global Change and Biodiversity of the University of Zurich. A.B. acknowledges the funding received from INTERACT (grant agreement no. 262693), under the European Community's Seventh Framework Programme. L.H., L.S.C., E.L., and A.T. acknowledge funding from NSERC-NCE ArcticNet. Any use of trade, firm, or product names is for descriptive purposes only and does not imply endorsement by the U.S. Government.

## Author contributions

M.G.C. conceived the study together with I.H.M.-S. and S.N. A.D.B. provided the abundance data and modelled cover change over time. A.B.-O. and S.N. created and provided the current and projected range data through Species Distribution Models. M.G.C. conducted the analyses and led the writing of the manuscript. I.H.M.-S., A.D.B., S.N., A.B.-O., H.J.D.T., A.E., K.H., J.M.A., A.A.-R., I.A., M.T.B., K.R.B.-M., D.B., A.B., B.E.L.C., K.C., J.H.C.C., B.C.F., E.R.F., P.G., L.H., R.D.H., J.H., M.I.-G., E.K., M.K., L.J.L., J.J.L., E.L., M.L., P.M., J.L.M., J.S.P., G.S.-S., S.N.S., L.S.C., N.A.S., A.T., S.E.V., and A.-M.V. provided trait data and/or contributed to the manuscript writing.

## Competing interests

The authors declare no competing interests.

## Additional information

[1]School of GeoSciences, University of Edinburgh, Edinburgh, Scotland, UK. [2]Department of Biology and Environmental Sciences, University of Gothenburg, Gothenburg, Sweden. [3]Gothenburg Global Biodiversity Centre, Gothenburg, Sweden. [4]Department of Biology, Aarhus University, Aarhus, Denmark. [5]Department of Physiological Diversity, Helmholtz Centre for Environmental Research - UFZ, Leipzig, Germany. [6]German Centre for Integrative Biodiversity Research (iDiv) Halle-Jena-Leipzig, Leipzig, Germany. [7]Department of Ecology and Genetics, University of Oulu, Oulu, Finland. [8]Environmental Science Center, Qatar University, Doha, Qatar. [9]CREAF, Cerdanyola del Vallès, Barcelona, Catalonia, Spain. [10]Institute of Botany and Landscape Ecology, University of Greifswald, Greifswald, Germany. [11]Natural Resources Canada, Canadian Forest Service, Great Lakes Forestry Centre, Sault Ste Marie, ON, Canada. [12]Copernicus Institute of Sustainable Development, Utrecht University, Utrecht, the Netherlands. [13]Centre for African Conservation Ecology, Nelson Mandela University, Port Elizabeth, South Africa. [14]Biology Department, Grand Valley State University, Allendale, MI, USA. [15]Dutch Research Council (NWO), The Hague, The Netherlands. [16]Land Surface-Atmosphere Interactions, School of Life Sciences Weihenstephan, Freising, Germany. [17]Department of Biotechnologies and Life Sciences, University of Insubria, Varese, Italy. [18]Threatened, Endangered, and Diversity Program, Alaska Department of Fish and Game, Anchorage, USA. [19]Section Systems Ecology, Amsterdam Institute for Life and Environment (A-LIFE), Vrije Universiteit, Amsterdam, The Netherlands. [20]Arctic Centre, University of Lapland, Rovaniemi, Finland. [21]WSL Institute for Snow and Avalanche Research SLF, Davos, Switzerland. [22]Swiss Federal Research Institute WSL, Birmensdorf, Switzerland. [23]Department of Geography, University of British Columbia, Vancouver, BC, Canada. [24]Climate Change and Extremes in Alpine Regions Research Centre CERC, Davos, Switzerland. [25]Department of Biology, Queen's University, Kingston, Ontario, ON, Canada. [26]Department of Biology, Memorial University, St. John's, NL, Canada. [27]Government of British Columbia, Vancouver, BC, Canada. [28]Department of Chemical and Biological Metrology, Federal Institute of Metrology METAS, Bern-Wabern, Switzerland. [29]Research Centre for Ecological Change, Organismal and Evolutionary Biology Research Programme, University of Helsinki, Helsinki, Finland. [30]Institute of Biology and Environmental Sciences, University of Oldenburg, Oldenburg, Germany. [31]Département des Sciences de l'environnement et Centre d'études nordiques, Université du Québec à Trois-Rivières, Trois-

Rivières, Québec, Canada. ³²Research Group Plants and Ecosystems (PLECO), University of Antwerp, Wilrijk, Belgium. ³³Department of Geosciences and Geography, University of Helsinki, Helsinki, Finland. ³⁴Institute of Hydrobiology, Biology Centre of the Czech Academy of Sciences, Ceske Budejovice, Czech Republic. ³⁵Department of Biological Sciences, Florida International University, Miami, FL, USA. ³⁶Department of Biology and Environmental Science, Marietta College, Marietta, OH, USA. ³⁷U.S. Geological Survey, Fort Collins, CO, USA. ³⁸Department of Evolutionary Biology and Environmental Studies, University of Zurich, Zurich, Switzerland. ³⁹Komarov Botanical Institute, St. Petersburg, Russia. ⁴⁰Terra Nova National Park, Parks Canada Agency, Glovertown, NL, Canada. ⁴¹Centre for Environmental Sciences, Hasselt University, Hasselt, Belgium. ⁴²School of Environment, Resources and Sustainability, University of Waterloo, Waterloo, ON, Canada. ⁴³Centre for Integrative Ecology, School of Life and Environmental Sciences, Deakin University, Burwood, VIC, Australia. ⁴⁴Woodwell Climate Research Center, Falmouth, MA, USA. ✉e-mail: mariana.garcia.criado@gmail.com

