## [Peer Review File · Nature Communications]

Plant traits poorly predict winner and loser shrub species in a warming tundra biomeREVIEWER COMMENTS

Reviewer #1 (Remarks to the Author):

This study assessed whether functional traits, in particular median trait values (MTV) and intraspecific trait variation (ITV), are associated with current range sizes and forecasted range shifts in tundra shrubs. In particular addressed the following questions: can traits explain current shrub species range size?, Do traits correspond with projected shrub range shifts and past cover change?, Which are the winner and loser shrub species in a warming tundra and what are their trait combinations. For this, authors gathered data for plant height, seed mass and specific leaf area for 62 shrub species inhabiting the arctic tundra, as well as geographical presence and climatic variables to characterize the current range of these species, and through modeling determine future potential range shifts. In general, the manuscript is well written, with clear objectives and addressing a relevant topic. Although I'm very sympathetic with this study, I do have some concerns about its novelty or further insights provided after the study of Bjorkman et al. published in Nature (2018). That study already addressed the question of the functional trait changes observed over the past three decades on plants at the arctic, including shrubs. Further, the study of Henn et al. (2018: Intraspecific trait variation and phenotypic plasticity mediate alpine plant species responses to climate change published in Frontiers in Plant Science) also addressed some of the topic tackled in this study. Thus, an effort to differentiate from previous contributions is badly needed. Maybe the part relating the relationship between traits and forecasted range shifts sounds to me more novel, although a thorough literature revision is needed.

My other concern is related with the traits used for the questions addressed. As authors properly acknowledged in the discussion section, they already know that, except for plant eight, the other two traits used are poorly related with the responses to climate change in tundra plant communities. Thus, it seems weird to address questions whose response is already anticipated.

I think that this work has a lot of potential, but I encourage authors to make an effort to work more on the novel parts in order to have a more compelling history

Reviewer #2 (Remarks to the Author):

This is an interesting, timely and novel study investigating trait signals in observed past plant cover changes and future projected range changes of tundra shrub species. The authors combine different indices of past and future range changes with an extensive trait data base and analyse how cover/range change relate to median traits and to trait variability. Although very interesting and with high potential, I have several comments for improving clarity and presentation, and streamlining the introduction and discussion.

Abstract: it should be mentioned that projected range shifts were quantified using SDMs.

Introduction: the introduction is lengthy and the structure could be improved. L 145 mentions intraspecific variation but the importance is not entirely clear. ITV is mentioned again in L 187-195 emphasising the persistent trait variation in tundra plants. Yet, in the following paragraph (> L 196) it is not motivated how ITV could be important or leave a signature. The potential meaning of ITV or how it is considered in this study only appears in the hypotheses starting in L 249. Here, the chain of argumentation could be improved and the introduction could be better streamlined. I am also wondering why only the amount of variation (sd of trait) is being considered rather than the extremes of the trait variation. Is that common practice, or is this strongly founded in ecological theory? Could it also be interesting to look at trait relationships with extremes?, e.g. studying the relationship between MTV and range size and between 5%/95%-quantile of TV and range size?

The introduction could also better motivate the difference between studying trait signals in past abundance/cover changes and in future range changes by SDM. Past abundance changes carry signatures of the changing environment but also of the biological processes and transient species responses to change. Future range shifts predicted by SDMs build on environmental descriptors of current ranges and ignore any transient dynamics. The latter is partly acknowledged in L 236-238. But then, the same hypotheses are suggested for both past and projected changes. Is this justified? The difference between observed cover change and predicted range change is discussed in L 531-536, showing that the authors are aware of that issue. But while this is acknowledged in parts of the discussion, it is poorly motivated in the introduction, is not reflected in hypotheses and results, and is also not acknowledged everywhere in discussion. For example, L 577-581 states that animal dispersion does not seem to have an effect on predicted range changes. But why should it, given that projections are only based on SDMs and not on explicitly modelled dispersal processes? Also, L 590-594 suggests that results indicate that future ITV benefits future range expansion because of plasticity. But why should it, given that projections are only based on SDMs and do not consider plasticity? Here, correlation should not be confused with causation.

Figure 1 also mentions dispersal rates that have not been mentioned in the introduction. L 615-618 mention for the first time that SDM predictions were constrained by migration ability.

Results: The analyses do not seem to distinguish between past and future winners/losers (L 324-330). If that is the case, how are winners and losers defined? If on the other hand, the authors did distinguish past and future winners/losers, then also the results should be reported separately. Fig. 3 also refers to top winners and losers, but it is not clear whether that refers to past or future. As L 347-349 refer to future winners/losers, it seems that previous results and Fig. 3 could relate to past winners/losers, but this is not made explicit anywhere and needs clarification. Also, in L 427-454 it is not clear what the winners and losers refer to, past or future. But then, L 456 mentions cover change and from that I could assume that the previous paragraphs only referred to future winners and losers. These are just a few examples. The main point here is that it is very confusing whether and when the authors distinguish between past and future winners/losers, and the text needs clarification and revision, as do the Figure captions.

The results also refer to additional categorical traits, e.g. dispersal mode and deciduousness, which are not motivated. The results text should at least explicitly state that additional to the three main continuous traits the authors tested for signals of these additional traits because they assumed that ...

Also, the discussion is very lengthy making it difficult to distil the main take home messages.

All figures are of very low quality; I hope this is just a problem that arose while rendering the pdf. Figure captions are not standalone; they should at least mention the study species (tundra shrubs) and location (tundra biomes).

Line comments:

L 163: The phrase "processes derived by climate change" is a bit awkward. Either edit to "induced by climate change" or rephrase e.g. to "mediate by climate change effects including ..."

L 664: edit "across the tundra biomes" as the study does not only consider Arctic?

REVIEWER COMMENTS

Reviewer #1 (Remarks to the Author):

This study assessed whether functional traits, in particular median trait values (MTV) and intraspecific trait variation (ITV), are associated with current range sizes and forecasted range shifts in tundra shrubs. In particular addressed the following questions: can traits explain current shrub species range size?, Do traits correspond with projected shrub range shifts and past cover change?, Which are the winner and loser shrub species in a warming tundra and what are their trait combinations. For this, authors gathered data for plant height, seed mass and specific leaf area for 62 shrub species inhabiting the arctic tundra, as well as geographical presence and climatic variables to characterize the current range of these species, and through modeling determine future potential range shifts.

In general, the manuscript is well written, with clear objectives and addressing a relevant topic. Although I'm very sympathetic with this study, I do have some concerns about its novelty or further insights provided after the study of Bjorkman et al. published in Nature (2018). That study already addressed the question of the functional trait changes observed over the past three decades on plants at the arctic, including shrubs. Further, the study of Henn et al. (2018: Intraspecific trait variation and phenotypic plasticity mediate alpine plant species responses to climate change published in Frontiers in Plant Science) also addressed some of the topic tackled in this study. Thus, an effort to differentiate from previous contributions is badly needed. Maybe the part relating the relationship between traits and forecasted range shifts sounds to me more novel, although a thorough literature revision is needed.

We thank the reviewer for their positive comments and we believe that their suggestions have thoroughly improved the manuscript.

We believe that there are sufficient novel elements to differentiate our study from the previously mentioned articles. The paper by Bjorkman et al. (2018) was indeed a ground-breaking study examining, however, solely changes in trait values over time. Our manuscript goes well beyond that by considering both trait values and intraspecific trait variation to relate to past cover changes, current range sizes, and also future projected range shifts. Henn et al. (2018) considered the effects of trait plasticity on changes in trait community values, but not on species population dynamics such as cover changes and projected range shifts. Additionally, Henn et al. (2018) focused on alpine plants at four sites in the Hengduan Mountains in China through an experimental approach which employed warming and cooling and reciprocal transplants. Meanwhile, our study focuses on the whole of the Arctic and relates trait data with observed ambient changes (with no experimental manipulations) and SDM-projected range shifts. Our study follows on from these and previous plant trait studies, and goes considerably further by relating said traits to plant community dynamics and range shifts.

We have mentioned Henn's study in the Introduction:

"Trait plasticity influences on trait community values has been assessed in site-level experiments⁵⁰, but has not considered population dynamics at the pan-Arctic scale." (lines 190-192)

Bjorkman's article is referenced throughout the Introduction:

"Community-level trait shifts have already been observed, with taller species spreading in a

*warming Arctic*¹².” (lines 133-134)

“In the warming tundra biome, community composition^{5,66,67} and certain size-related and resource economics traits are changing across time and space^{12,42}.” (lines 232-234)

“Tundra plants occurring in warmer climates tend to have greater height and SLA^{12,40,42}.” (lines 271-272)

We now better emphasised the novelty of our study by rephrasing the introductory pitch for the study:

“However, the explanatory power of plant traits on species’ past cover change, their current range size or and their potential for future range shifts across the Arctic remains unknown. have not been explored.” (lines 234-236).

“To overcome this knowledge gap, we combine species trait, range and abundance data to understand whether median trait values (MTV) and intraspecific trait variation (ITV) are associated with current range sizes in tundra shrubs.” (lines 246-248)

We also emphasise the novelty of our findings in the conclusions section:

“Our findings indicate that no specific combination of trait values or variation is associated with winner or loser tundra shrub species under climate change. Contrary to our expectations, particular trait values or greater trait variation do not necessarily indicate increased range or abundance shifts, although there was a broadly positive signal of greater seed mass values with projected range shifts, and greater SLA and seed mass variation with projected range shifts.” (lines 722-727)

My other concern is related with the traits used for the questions addressed. As authors properly acknowledged in the discussion section, they already know that, except for plant eight, the other two traits used are poorly related with the responses to climate change in tundra plant communities. Thus, it seems weird to address questions whose response is already anticipated.

We understand the comment and hope that our edits have more clearly stated the novelty of the results (see comment above). Going into the study we expected all three traits to be associated with past cover change, current ranges and projected range shifts given their documented relationships with species performances and ecosystem function (as indicated in the introduction), and the fact that many of the main species expanding quickly across the Arctic have similar favourable traits in common. Just as an example, *Salix* willows have very small seeds with high dispersal capacity, and thus one of our hypotheses was that seed mass would be a relevant predictor of population dynamics through lighter seeds reaching farther distances. Thus, we were greatly surprised that the explanatory power of these traits was so low, as we were expecting to find a strong relationship for all traits.

We discuss these results in the Discussion section and put them in the context of the literature that found similar results, but many other articles with other databases and ecosystems have shown better explanatory power of traits on range dynamics and ecosystem function (e.g., Sporbert et al. 2021, Hagan et al. 2023). We make reference to this in the Discussion:

“While traits have been extensively linked to predicting range dynamics and ecosystem

function^{80,81}, we found that the traits used in this study were weakly related to the projected range shifts and past cover change of tundra shrubs (Figure 5)." (lines 577-580)

We have also rephrased accordingly further in the Discussion:

"Plant height, SLA and seed mass are response traits which should influence species' abilities of persisting in and colonise changing habitats⁷⁰. Contrary to our expectations, range winners tended to have heavier seeds than loser and no change species. Although plants with lighter seeds tend to disperse further via wind and produce more seeds^{52,71}, larger seeds are more likely to be found within berries that are dispersed by animals over longer distances⁷², and are advantageous for seedling establishment due to more storage tissue^{52,73}." (lines 524-535)

I think that this work has a lot of potential, but I encourage authors to make an effort to work more on the novel parts in order to have a more compelling history

We thank the reviewer for his very useful feedback and hope that our changes, as noted above, have made the novelty of our study more evident.

Reviewer #2 (Remarks to the Author):

This is an interesting, timely and novel study investigating trait signals in observed past plant cover changes and future projected range changes of tundra shrub species. The authors combine different indices of past and future range changes with an extensive trait database and analyse how cover/range change relate to median traits and to trait variability. Although very interesting and with high potential, I have several comments for improving clarity and presentation, and streamlining the introduction and discussion.

Thank you very much for your thorough review and the very useful suggestions. We have implemented these and believe they have greatly improved the manuscript.

Abstract: it should be mentioned that projected range shifts were quantified using SDMs.

We agree and have now incorporated this comment into the Abstract, which now reads as follows:

"Here, we investigate whether past abundance changes, current range sizes and projected range shifts derived from species distribution models are related to plant trait values and intraspecific trait variation. We combined 17,921 trait records with observed past and modelled future distributions from 62 shrub species across three continents." (lines 76-80)

Introduction: the introduction is lengthy and the structure could be improved. L 145 mentions intraspecific variation but the importance is not entirely clear. ITV is mentioned again in L 187-195 emphasising the persistent trait variation in tundra plants. Yet, in the following paragraph (> L 196) it is not motivated how ITV could be important or leave a signature. The potential meaning of ITV or how it is considered in this study only appears in the hypotheses starting in L 249. Here, the chain of argumentation could be improved and the introduction could be better streamlined.

We have now shortened and reorganised the introduction section by removing one paragraph and improving the logic behind the importance of ITV. This content was previously in the hypothesis section but we agree it makes much more sense to already

cover this point in the introduction:

“Thus, intraspecific trait variation (ITV) might have a strong influence on ecological dynamics^{41,47}. ITV accounts for 25% of total trait variation within communities, 32% among communities⁴⁵, and 23% of trait variation in tundra biome-wide data⁴². Indeed, ITV is an important component of environmental matching, and greater ITV via genetic or phenotypic variation could provide more opportunities for natural selection and adaptation⁴¹, increasing species’ chances of adapting to fluctuating environmental conditions^{48,49}.” (lines 184-190)

We have also re-worded the hypothesis accordingly:

“We hypothesise that greater ITV in all three traits would be positively related to species’ range sizes, reflecting greater adaptations to environmental variability^{41,48}.” (lines 263-265)

I am also wondering why only the amount of variation (sd of trait) is being considered rather than the extremes of the trait variation. Is that common practice, or is this strongly founded in ecological theory? Could it also be interesting to look at trait relationships with extremes?, e.g. studying the relationship between MTV and range size and between 5%/95%-quantile of TV and range size?

The standard deviation (SD) is indeed one of the most commonly used metrics to measure intraspecific trait variation, as it is easily comparable across species and allows us to quantify the extent of the amount of variation present in populations (Mitchell and Bakker 2013, Westerland et al. 2021). The SD encompasses the full variation in trait values, whereas the extremes only represent the tails of the trait distribution. We thus believe that looking at SD is more meaningful in our analyses.

Our original hypotheses were that species with greater ITV would be better placed to expand their ranges with current and future warming, as ITV could reflect greater adaptation capacity and possibility of coping with natural selection. Thus, using SD as our ITV metric allowed us to reflect this genotypic and phenotypic variation and to understand its links with projected range shifts, current ranges, and past abundance changes.

It would be very interesting indeed to examine trait relationships with minimum and maximum extreme trait values, and this has not been widely explored in the literature (Yang et al. 2020). However, using min-max metrics poses a set of different questions and hypotheses to those that we posited in this study, and would require running the equivalent of over 150 models with min-max metrics (Table S2).

Additionally, extreme trait values are more sensitive to sampling bias, as we would need at least 30 trait records per species to be confident that we are capturing the tails of each trait distribution accurately. Thus, we would need additional records to be able to have confidence in our results if using min-max values. However, we agree that examining the trait-range relationship using extreme trait values would be very interesting, particularly with a database where traits are better sampled at the extremes.

We refer to our choice of SD as an ITV metric in the Methods section:

“We chose SD as a commonly used ITV metric with a more conservative data distribution than others like the coefficient of variation (COV); however both metrics were directly proportional (Table S4-S6, Figure S6).” (lines 774-778)

We have also included a sentence about using different metrics of trait variation in the Discussion:

“Additionally, analysing different metrics of trait variation renders an interesting research avenue but requires larger sample networks to ensure robustness of results, and thus should be considered in future studies.” (lines 700-702)

The introduction could also better motivate the difference between studying trait signals in past abundance/cover changes and in future range changes by SDM. Past abundance changes carry signatures of the changing environment but also of the biological processes and transient species responses to change. Future range shifts predicted by SDMs build on environmental descriptors of current ranges and ignore any transient dynamics. The latter is partly acknowledged in L 236-238.

We have addressed these points in the introduction as follows:

“Thus, range projections cannot fully mirror future species distributions in the same way as long-term observations could, which reflect not only changes in the environment, but also the effect of biological processes and transient species responses^{58,59}.” (lines 220-223)

“However, the explanatory power of plant traits on species’ past cover change, their current range size and their potential for future range shifts across the Arctic remain unknown.” (lines 234-236)

But then, the same hypotheses are suggested for both past and projected changes. Is this justified? The difference between observed cover change and predicted range change is discussed in L 531-536, showing that the authors are aware of that issue. But while this is acknowledged in parts of the discussion, it is poorly motivated in the introduction, is not reflected in hypotheses and results, and is also not acknowledged everywhere in discussion. For example, L 577-581 states that animal dispersion does not seem to have an effect on predicted range changes. But why should it, given that projections are only based on SDMs and not on explicitly modelled dispersal processes? Also, L 590-594 suggests that results indicate that future ITV benefits future range expansion because of plasticity. But why should it, given that projections are only based on SDMs and do not consider plasticity? Here, correlation should not be confused with causation.

We can see how our same hypotheses for both past and projected changes could indeed be seen as simplistic, which is why we address this in the Discussion (see ‘Moving beyond functional traits’ section starting line 621). However, these were our original hypotheses when we started this work, i.e., that individuals occupying warmer climatic niches and having more competitive strategies and increased dispersal capacity will occupy larger projected ranges and have undergone cover increases. As can be seen in the introduction, there is sufficient foundation in the literature for these hypotheses to make sense. We therefore believe that we should be truthful to our evidence-based expectations and leave the hypotheses as they are, while emphasising why these initial expectations were not met in the discussion.

When we initiated this work, we saw the two metrics (past abundance and projected ranges) as different timeframes of the same process (e.g., species-level shifts) and thus had similar expectations for the effect of traits, considering that abundance and ranges are usually related following abundance-range theory (Gaston and Blackburn 2008). Nevertheless, we have tweaked the hypothesis to reflect the reviewer’s comment:

“With projected warming¹, we expect that species occupying warmer climatic niches and

having more competitive strategies (greater height and SLA values) and increased dispersal capacity (small seeds) will occupy larger projected ranges and will have undergone cover increases under a warming climate, despite past abundance changes reflecting species responses in a way that projections cannot.” (lines 272-278)

We have also emphasised this point in the Results:

“Only 10 species shared the same range and cover categories (i.e., are consistently either winners, losers or no change species in both methods), with four winners, one no change and five loser species in common, showing that the abundance method reflects biological processes and species responses that range projections cannot.” (Lines 386-390)

We have also emphasised the point that our hypotheses were based on previous literature results:

“Our initial hypothesis, based on previous literature, of tundra shrubs showing consistent trait responses to climate change turned out to be too simplistic.” (lines 622-623)

With regards to the animal-dispersal point, our SDM projections did actually incorporate species-specific migration rates (see “Supplementary Methods – Species Distribution Models (SDMs) - Migration rates”), thus we believe the chosen wording is still relevant to reflect the effect of dispersal on range projections. However, we take the reviewer’s point about ITV not being included in SDMs, and our intention was to indicate that those species that were projected to expand the most had greater ITV, and not that having greater ITV was the cause of expanding the most. We have re-worded accordingly:

“We found an indication that species projected to expand the most had greater SLA and seed mass variation, suggesting that winner species could be more plastic or adaptable⁴¹.” (lines 613-616)

Figure 1 also mentions dispersal rates that have not been mentioned in the introduction. L 615-618 mention for the first time that SDM predictions were constrained by migration ability.

Indeed, the SDM projections were calculated considering species-specific dispersal capabilities. This is explained in the Methods section ‘Range size data’ (starting in line 829), and the SDM modelling is expanded in more detail in the Appendix section ‘Supplementary Methods - Species Distribution Models (SDMs)’. We have now mentioned the dispersal rates in the Introduction as well:

“We determine which categorical and continuous traits are associated with species projected from SDMs featuring dispersal rates to expand or decrease their ranges, and with species that have increased or decreased in abundance over time (Figure 1).” (lines 252-255)

Results: The analyses do not seem to distinguish between past and future winners/losers (L 324-330). If that is the case, how are winners and losers defined? If on the other hand, the authors did distinguish past and future winners/losers, then also the results should be reported separately. Fig. 3 also refers to top winners and losers, but it is not clear whether that refers to past or future. As L 347-349 refer to future winners/losers, it seems that previous results and Fig. 3 could relate to past winners/losers, but this is not made explicit anywhere and needs clarification. Also, in L 427-454 it is not clear what the winners and losers refer to, past or future. But then, L 456 mentions cover change and from that I could assume that the previous paragraphs only referred to future winners and losers. These are

just a few examples. The main point here is that it is very confusing whether and when the authors distinguish between past and future winners/losers, and the text needs clarification and revision, as do the Figure captions.

We identified winners and losers through two different methods as indicated in the Methods section 'Classification of winner and loser shrub species' (starting in line 866). We have edited this section to better signpost that 'range' winners/losers refer to future projections and 'cover' winners/losers refer to past changes:

"We classified winner and loser shrub species using 1) projected range shifts from the SDMs (into the years 2070 – 2099) and 2) cover change over time from the International Tundra Experiment (ITEX) dataset (between 1970 - 2010). For range shift projections, we calculated the 25%, 50% and 75% quantiles of species' projected range shifts across the 24 climatic scenarios (both for absolute and relative range shifts) and categorised species as projected future 'range winners' (if the 25% quantile was above zero), no change (if any quantile overlapped zero) or projected future 'range losers' (if the 75% quantile was below zero). For cover change over time, we analysed shrub cover change over time from 105 subsites and 30 sites from the ITEX network (Henry and Molau 1997). Based on the analysis by Bjorkman et al. (2018), individual species' relative cover change over time per plot were modelled as ordinal numbers using a Poisson distribution with subsite and site as random effects, aggregating after to subsite and species level. Thus, we obtained slopes of cover change per year for each species-by-site combination. We defined past cover winner, no change and loser categories according to whether these slopes per species across all sites were positive or negative, and whether the 95% credible intervals overlapped zero (Figure S3, Table S3)." (lines 867-883)

We describe these in the text as "range winners/losers" and "cover winners/losers". We have gone through the text and made sure to specify which group of winner/loser we're referring to all throughout the text, including figure captions. We left winner/loser mentions without specification only when we referred to winner/losers in general as a concept, or when encompassing both range and cover winner and losers. We have also specified that 'cover winners' refer to an assessment method of past cover changes, and 'range winners' refer to future projected distributions in key sections of the text:

"Finally, we presume that species that have increased in cover (i.e., past 'cover winners') are also projected to experience range expansions with warming (i.e., projected future 'range winners'), following the abundance-range size relationship theory⁶⁸." (lines 294-297)

"In this study, projected future range winner species were more likely to have greater seed mass values, and greater variation in SLA and seed mass compared to losers, potentially conferring an advantage in a warmer future climate." (lines 509-512)

"Additionally, winner and loser species differ when comparing past cover change over time with future projected range shifts." (lines 731-732)

The results also refer to additional categorical traits, e.g. dispersal mode and deciduousness, which are not motivated. The results text should at least explicitly state that additional to the three main continuous traits the authors tested for signals of these additional traits because they assumed that ...

We have now introduced the categorical traits earlier in the Introduction:

"Additional categorical traits with potential to influence species dynamics are dispersal mode, deciduousness, functional group and taxonomy". (lines 209-210)

We have included specific hypotheses for the categorical traits:

“We expect deciduous and wind-dispersed species from Salicaceae and Betulaceae families to be winners due to their rapid resource acquisition, long-distance dispersal and flexible colonisation strategies.” (lines 281-283)

We have also signposted more the categorical results in the Results section:

“In contrast with our hypotheses, there were no differences in current range sizes depending on categorical traits, including species’ dispersal mode, deciduousness, functional group or taxonomic family, except for Salicaceae species having smaller ranges than species from the Rosaceae family (Table S2.29).” (lines 342-346)

“There were no differences in projected range shifts depending on the categorical traits.” (lines 426-427)

Also, the discussion is very lengthy making it difficult to distil the main take home messages. We have now shortened the Discussion by removing one paragraph and considerably shortening two other paragraphs. We believe that the main take-home messages are clearer now.

All figures are of very low quality; I hope this is just a problem that arose while rendering the pdf.

We have now uploaded all the main figures individually and with high quality and hope this is reflected in the figure quality after PDF rendering. All figures have been created in a .PNG format with a 500x500 dpi resolution, well above the designing standard for a good resolution of 300x300 dpi.

Figure captions are not standalone; they should at least mention the study species (tundra shrubs) and location (tundra biomes).

We have edited all manuscript and appendix figure captions to stand alone. In doing so, we also specify that we refer to tundra shrubs. We have also spelled out the acronyms MTV (mean trait values) and ITV (intraspecific trait variation) in Figures 3, 5 and 6.

Line comments:

L 163: The phrase “processes derived by climate change” is a bit awkward. Either edit to “induced by climate change” or rephrase e.g. to “mediate by climate change effects including ...”

Corrected. We have rephrased this sentence:

“Current range shifts are mediated by processes induced by climate change, including permafrost thaw, earlier snow melt, extended season length, increased nutrient availability and species interactions²⁶.” (lines 161-164)

L 664: edit “across the tundra biomes” as the study does not only consider Arctic?

Corrected. We have rephrased this sentence:

“Further trait data collection across the tundra biome over time would enable the replication of these analyses based on a larger number of morpho-physiological traits and species.”
(lines 678-680)

References included in this document

Bjorkman, A. D. *et al.* (2018). Plant functional trait change across a warming tundra biome. *Nature* **562**, 57.

Gaston, K. J., & Blackburn, T. M. (2008). *Pattern and Process in Macroecology*. Blackwell Publishing.

Hagan, J. G., *et al.* (2023). Plant traits alone are good predictors of ecosystem properties when used carefully. *Nature Ecology & Evolution*, 1-3.

Henn, J. J. *et al.* Intraspecific Trait Variation and Phenotypic Plasticity Mediate Alpine Plant Species Response to Climate Change. *Frontiers in Plant Science* **9**, (2018).

Mitchell, R. M., & Bakker, J. D. (2014). Quantifying and comparing intraspecific functional trait variability: a case study with *Hypochaeris radicata*. *Functional Ecology*, 28(1), 258-269.

Sporbert, M., *et al.* (2021). Different sets of traits explain abundance and distribution patterns of European plants at different spatial scales. *Journal of Vegetation Science*, 32(2), e13016.

Yang, J., *et al.* (2020). Large underestimation of intraspecific trait variation and its improvements. *Frontiers in Plant Science*, 11, 53.

Westerband, A. C *et al.* (2021). Intraspecific trait variation in plants: a renewed focus on its role in ecological processes. *Annals of botany*, 127(4), 397-410.

REVIEWERS' COMMENTS

Reviewer #2 (Remarks to the Author):

First, please accept my apologies for the delay in this review due to unusually hectic time of the year. This is the second time I review this manuscript and I congratulate and thank the authors for this fine revision. The clarifications in the response letter and manuscript are very convincing and helpful.

The introduction and discussion is much better streamlined, and the study provides an interesting read. I still have a couple of minor comments left:

In my previous review, I stressed that future range projections based on SDMs do not carry any signatures of transient ecological dynamics and it is thus a bit farther fetched to discuss the effect of dispersal and competitive traits on range winners/losers. Although I largely agree with the authors' response to this, I still suggest slight amendments to the discussion. Mainly, in L 523ff (section "Winners and losers"), it would be good to acknowledge that the mentioned processes (wind dispersal, seedling establishment, competitive advantage) manifest themselves in the realised niche and here the authors project this potential realised niche into the future. Thus, to some extent these projections may carry some signatures of these processes.

I suggest considering to use the terms "past winners/losers) and "future winners/losers" rather than "cover w/l" and "range w/l", and may make it slightly easier to read. This is simply a suggestion and I would not press this point. But I do think that it would improve readability. This point concerns the entire results section where most text simply refers to cover changes or range shifts.

Figure 2: I am wondering whether this graph (esp. panel b) would not be better placed in appendix while only showing a more summarised plot in main text. For the main text, the identity of the different species is less important. The main information in this graph is rather the trait distribution across species and within species. Again, this is simply a suggestion but I would not press this point.

Line comments:

- L 129: remove one "to" in "up to to four times"
- L 165: remove "their" in "species' their colonisation"
- L 389-390: "showing that the abundance method reflects biological processes and species responses that range projections cannot" – I would further differentiate this points. First of all, it shows that observed past and modelled future winners and losers are not necessarily identical. And only in the second moment, the future projections do not carry the same signature of biological processes.
- L 508-509: please slightly rephrase this opening sentence of the discussion. As currently written, it is not entirely clear whether this refer to results of the present study or to a general statement. It seems to be meant as general statement.
- L 677: "TRY/TTT" should be spelled out at first usage.
- Harmonise spelling between British/American English, e.g. both "colonization" and "colonisation" left in the text

REVIEWERS' COMMENTS

Reviewer #2 (Remarks to the Author):

First, please accept my apologies for the delay in this review due to unusually hectic time of the year. This is the second time I review this manuscript and I congratulate and thank the authors for this fine revision. The clarifications in the response letter and manuscript are very convincing and helpful. Thank you very much for your feedback and for assessing our manuscript once more. We have addressed all the comments below and our responses are in blue. References to manuscript lines refer to the version without track changes.

The introduction and discussion is much better streamlined, and the study provides an interesting read. I still have a couple of minor comments left:

In my previous review, I stressed that future range projections based on SDMs do not carry any signatures of transient ecological dynamics and it is thus a bit farther fetched to discuss the effect of dispersal and competitive traits on range winners/losers. Although I largely agree with the authors' response to this, I still suggest slight amendments to the discussion. Mainly, in L 523ff (section "Winners and losers"), it would be good to acknowledge that the mentioned processes (wind dispersal, seedling establishment, competitive advantage) manifest themselves in the realised niche and here the authors project this potential realised niche into the future. Thus, to some extent these projections may carry some signatures of these processes.

We have added the following sentence to the 'Winners and losers' section where we discuss the mismatch between the two different assessment methods (lines 386-388):

"While dispersal and establishment processes are manifested in realised niches, and thus in projections to a certain extent, transient ecological dynamics are not captured by future projections."

I suggest considering to use the terms "past winners/losers) and "future winners/losers" rather than "cover w/l" and "range w/l", and may make it slightly easier to read. This is simply a suggestion and I would not press this point. But I do think that it would improve readability. This point concerns the entire results section where most text simply refers to cover changes or range shifts.

We agree with this suggestion and have replaced "cover winners/losers" with "past winners/losers" and "range winners/losers" with "future winners/losers" throughout the document.

Figure 2: I am wondering whether this graph (esp. panel b) would not be better placed in appendix while only showing a more summarised plot in main text. For the main text, the identity of the different species is less important. The main information in this graph is rather the trait distribution across species and within species. Again, this is simply a suggestion but I would not press this point. Thank you for this suggestion. There is consensus across the co-authors that Figure 2 is quite informative for the reader as it clarifies the context of the study by showing data collection localities and species-level trait data, and it sets up the following graphs and analyses. Therefore, if the reviewer is happy for us to keep it in, we'd rather keep it in the main text.

Line comments:

- L 129: remove one "to" in "up to to four times".

Corrected.

- L 165: remove “their” in “species’ their colonisation”

Corrected.

- L 389-390: “showing that the abundance method reflects biological processes and species responses that range projections cannot” – I would further differentiate this points. First of all, it shows that observed past and modelled future winners and losers are not necessarily identical. And only in the second moment, the future projections do not carry the same signature of biological processes.

We have opted for deleting this statement considering that it was an interpretation of the results, and thus should be in the discussion and not in the results section. We believe that this point has been addressed as part of the reviewers’ comment above on transient dynamics, and that the discussion section in lines 377-394 is a more appropriate place to discuss this matter:

“While taller species represent more future winners than shorter species (Figure 6a), this climate-trait mismatch could mean that tall shrubs will not necessarily take over the landscape, as frequently reported in tundra projections. Surprisingly, only 10 of the 36 shrubs (27.7%) with data on past cover change over time shared the same winner/loser categories as the species range categories (Figure 4, Table S2, Figure S3). This result does not support the generally accepted abundance-range size theory⁶⁸, but agrees with other studies⁷⁸. A potential explanation is that the SDM-derived ranges identify potential future climatic niches constrained by boundaries set by species-specific migration rates, rather than the real-world climate responses of tundra shrubs. While dispersal and establishment processes are manifested in realised niches, and thus in projections to a certain extent, transient ecological dynamics are not captured by future projections. For instance, a species could be classified as a future winner because of an expanded climatic niche, but as a past loser because of decreased abundance, meaning that its fundamental niche does not track its potential future climatic niche. Conversely, a species may be classified as a loser because of a projected range contraction, but be able to persist in situ and adapt to changing climatic conditions, which SDM projections would not be able to capture.”

- L 508-509: please slightly rephrase this opening sentence of the discussion. As currently written, it is not entirely clear whether this refer to results of the present study or to a general statement. It seems to be meant as general statement.

Indeed, this opening sentence was meant as a general statement which sets up the related results in our study. Nevertheless, we have rephrased this as “Species’ range and abundance shifts are forecasted with climate change” (line 348), which is a bit shorter but still sets up the following results.

- L 677: “TRY/TTT” should be spelled out at first usage.

TTT (Tundra Trait Team) is indeed spelled out in lines 538-539. However, TRY is not an acronym, but rather the name of the database instead. From Kattge et al. (2011): “the TRY initiative (TRY – not an acronym, rather an expression of sentiment: <http://www.try-db.org>)”.

Kattge, J., Diaz, S., Lavorel, S., Prentice, I. C., Leadley, P., Bönsch, G., ... & Wirth, C. (2011). TRY—a global database of plant traits. *Global change biology*, 17(9), 2905-2935.

- Harmonise spelling between British/American English, e.g. both “colonization” and “colonisation” left in the text.

We have fixed the “colonization” in line 111 to the British version “colonisation”. We have scanned the text for this issue and also fixed “realization” to “realisation” in line 400.